# Single-cell eQTL mapping in yeast reveals a tradeoff between growth and reproduction

**James Boocock**[1,2,3], **Noah Alexander**[1,2,3], **Leslie Alamo Tapia**[1,2,3], **Laura Walter-McNeill**[1,2,3], **Shivani Prashant Patel**[1,2,3], **Chetan Munugala**[1,2,3], **Joshua S Bloom**[1,2,3]*, **Leonid Kruglyak**[1,2,3]*

[1]Department of Human Genetics, University of California, Los Angeles, Los Angeles, United States; [2]Department of Biological Chemistry, University of California, Los Angeles, Los Angeles, United States; [3]Howard Hughes Medical Institute, Chevy Chase, United States

*For correspondence:
JBloom@mednet.ucla.edu (JSB);
LKruglyak@mednet.ucla.edu (LK)

## eLife assessment

This manuscript describes the mapping of natural DNA sequence variants that affect gene expression and its noise, as well as cell cycle timing, using as input single-cell RNA-sequencing of progeny from crosses between wild yeast strains. The method represents an **important** advance in the study of natural genetic variation. The findings, especially given the follow-up validation of the phenotypic impact of a mapped locus of major effect, provide **convincing** support for the rigor and utility of the method.

**Abstract** Expression quantitative trait loci (eQTLs) provide a key bridge between noncoding DNA sequence variants and organismal traits. The effects of eQTLs can differ among tissues, cell types, and cellular states, but these differences are obscured by gene expression measurements in bulk populations. We developed a one-pot approach to map eQTLs in *Saccharomyces cerevisiae* by single-cell RNA sequencing (scRNA-seq) and applied it to over 100,000 single cells from three crosses. We used scRNA-seq data to genotype each cell, measure gene expression, and classify the cells by cell-cycle stage. We mapped thousands of local and distant eQTLs and identified interactions between eQTL effects and cell-cycle stages. We took advantage of single-cell expression information to identify hundreds of genes with allele-specific effects on expression noise. We used cell-cycle stage classification to map 20 loci that influence cell-cycle progression. One of these loci influenced the expression of genes involved in the mating response. We showed that the effects of this locus arise from a common variant (W82R) in the gene *GPA1*, which encodes a signaling protein that negatively regulates the mating pathway. The 82R allele increases mating efficiency at the cost of slower cell-cycle progression and is associated with a higher rate of outcrossing in nature. Our results provide a more granular picture of the effects of genetic variants on gene expression and downstream traits.

## Introduction

Genome-wide studies have identified thousands of loci that influence gene expression; these loci are known as expression quantitative trait loci or eQTLs (*Albert and Kruglyak, 2015*; *Kang et al., 2023*). eQTLs serve as an important bridge between DNA sequence variation and organismal phenotypes and provide a mechanism by which noncoding variants can underlie complex traits (*Finucane et al.,*

*2015*; *Umans et al., 2021*; *Gusev et al., 2016*). The vast majority of eQTL studies to date have relied on measurements of average gene expression levels in bulk populations of cells (*Aguet et al., 2020*; *Albert et al., 2018*). This approach, while experimentally tractable, can lose information about known differences in genetic effects among tissues (*Aguet et al., 2020*), cell types (*Westra et al., 2015*; *Chen et al., 2016*; *Kim-Hellmuth et al., 2020*; *Ota et al., 2021*), and cellular states (*Strober et al., 2019*). Recently, studies in humans have leveraged single-cell RNA sequencing (scRNA-seq) to more flexibly investigate how eQTL effects are altered in different contexts (*Elorbany et al., 2022*; *Cuomo et al., 2020*; *Neavin et al., 2021*; *Jerber et al., 2021*; *Yazar et al., 2022*; *van der Wijst et al., 2018*), including cellular states that are difficult to access with bulk approaches (*Nathan et al., 2022*). However, obtaining these more granular eQTL maps with either bulk or single-cell approaches comes at the cost of substantial increases in the numbers of samples that must be obtained and analyzed one at a time.

In model organisms, such as the nematode *Caenorhabditis elegans* and the budding yeast *Saccharomyces cerevisiae*, mapping populations of millions of recombinant progeny can be generated in a single flask (*Ehrenreich et al., 2010*; *Burga et al., 2019*). Such populations can be combined with scRNA-seq in a 'one-pot' eQTL mapping design in which the same single-cell data enables measurement of gene expression, cell type classification, and genotyping of transcribed variants in each cell (*Ben-David et al., 2021*). This design has two major advantages. First, it retains information about tissues, cell types, and cellular states. Second, by replacing expensive and labor-intensive genotyping and expression profiling of many samples with a single scRNA-seq experiment, it enables facile exploration of genetics of gene expression in many different genetic backgrounds and in response to many environmental perturbations. Here, we implement this design in yeast (*Figure 1A*), which presents additional challenges due to small cell size and the presence of a cell wall (*Gasch et al., 2017*; *Jariani et al., 2020*; *Nadal-Ribelles et al., 2019*; *Jackson et al., 2020*; *Brettner et al., 2022*; *N'Guessan et al., 2023*), and use it to identify eQTLs in different genetic backgrounds, study interactions between eQTL effects and stages of the cell cycle, search for allele-specific effects on gene expression noise, and uncover a connection between a common variant in the gene *GPA1*, gene expression, progression through the cell cycle, and mating efficiency.

## Results

### scRNA-seq enables simultaneous expression profiling, cell-cycle stage determination, and genotyping in a segregating yeast population

eQTL mapping requires tracking the inheritance of genetic variants and measuring gene expression in the same individuals. scRNA-seq captures the transcriptomes of individual cells, and genotypes of expressed single-nucleotide polymorphisms (SNPs) in transcribed sequences can be used to track inheritance in these same cells. We previously showed that this approach enables single-cell eQTL mapping in *C. elegans* (*Ben-David et al., 2021*). To test the feasibility of the approach in yeast, we pooled 393 previously genotyped haploid segregants from a cross between a lab strain (BY) and a wine strain (RM) (*Albert et al., 2018*; *Bloom et al., 2013*; *Figure 1—figure supplement 1*; *Supplementary file 1, tables S1–S3*) and used scRNA-seq to obtain the transcriptomes of 7124 cells (Methods, *Supplementary file 1, table S4*). We captured a median of 1514 unique RNA molecules (unique molecular identifiers; henceforth UMIs) and a median of 1091 expressed SNPs per cell (*Supplementary file 1, table S4*).

The expression of hundreds of yeast genes varies during progression through the stages of the cell cycle (*Spellman et al., 1998*). We classified individual haploid yeast cells into five different cell-cycle stages (M/G1, G1, G1/S, S, and G2/M) via unsupervised clustering of the expression of 787 cell-cycle-regulated genes (*Spellman et al., 1998*) in combination with 22 cell-cycle-informative marker genes (*Figure 1B*, *Figure 1—figure supplements 2–5*). Using this classification approach, we found that expression of 2139 genes displayed significant variation by cell-cycle stage (likelihood ratio test, false-discovery rate [FDR] <0.05; *Supplementary file 1, table S5*). To account for the observed widespread effects of the cell cycle on gene expression, we incorporated the cell-cycle stage into subsequent eQTL analyses, unless otherwise stated.

We used a hidden Markov model (HMM) to reconstruct the patterns of inheritance of parental alleles in each cell based on the observed genotypes at expressed SNPs. This allowed us to match

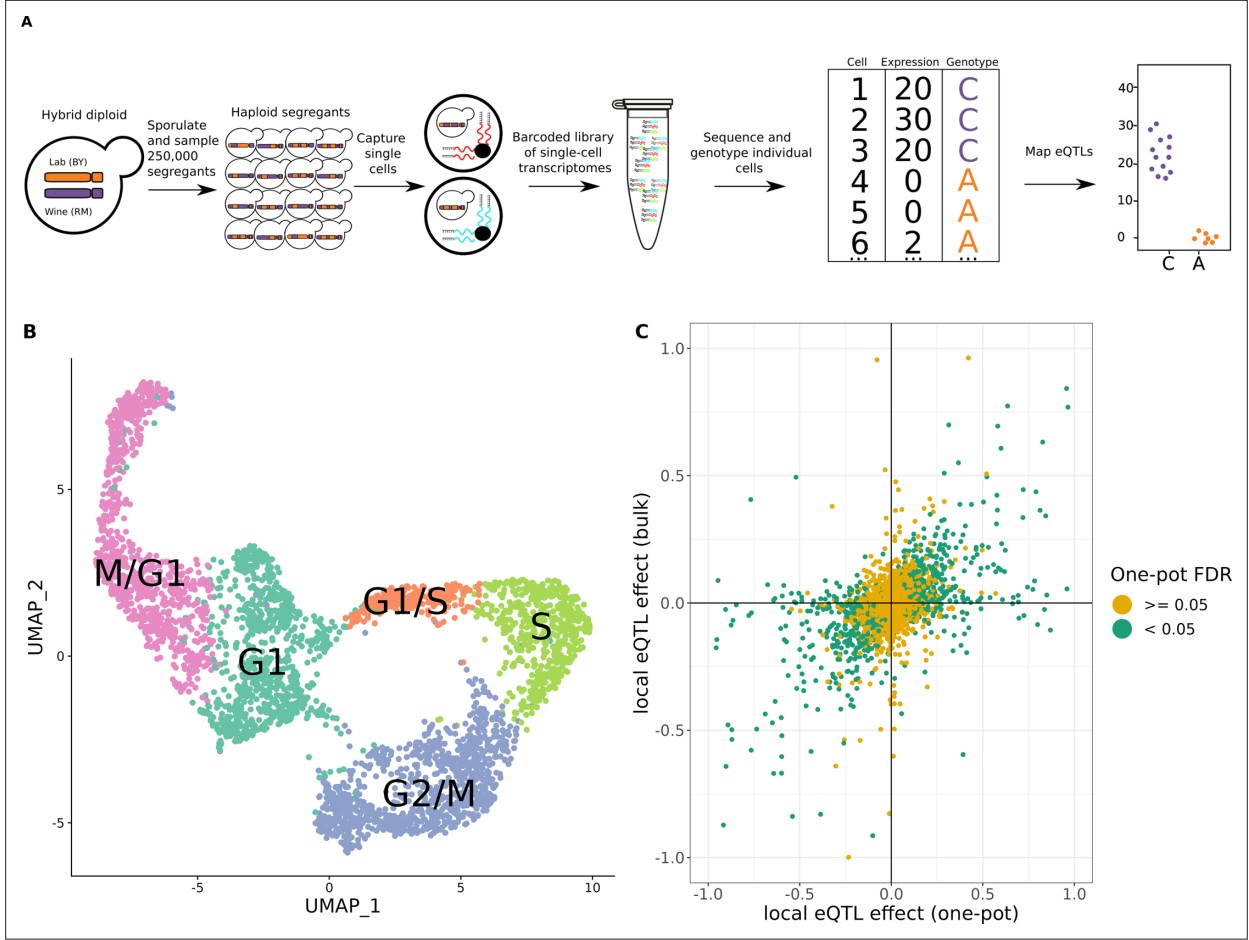

**Figure 1.** One-pot eQTL mapping is feasible in yeast. (**A**) One-pot eQTL mapping workflow. A large population of hybrid diploid cells is sporulated, and MATa haploid yeast progeny cells (segregants) are isolated by fluorescence-activated cell sorting. Cells are captured and processed with the 10× Chromium device. The resulting barcoded library of single-cell transcriptomes is sequenced by Illumina short-read sequencing. Unique molecular identifier (UMI) counts are tallied for each transcript in each segregant. The number of supporting molecules for each parental allele is identified at every transcribed sequence position that differs between the parental strains, and a hidden Markov model is used to infer the genotype of each segregant. In the cartoon example of an eQTL shown on the top right, segregants with the C allele have higher expression of the gene than those with the A allele. (**B**) Representative Uniform Manifold Approximation and Projection for Dimension Reduction (UMAP) plot of cells colored by their assigned cell-cycle stage. (**C**) Scatter plot of local eQTL effects from the one-pot experiment in the cross between BY and RM (*x*-axis) against local eQTL effects based on expression measurements from bulk RNA-seq in the same cross (*y*-axis) (*Albert et al., 2018*). Green dots denote one-pot eQTL effects that were significant at a false-discovery rate (FDR) of 0.05; yellow dots denote those that were not. The *x*- and *y*-axis were truncated at –1 and 1 for ease of visualization, which left out 67 of 4044 data points.

The online version of this article includes the following figure supplement(s) for figure 1:

**Figure supplement 1.** Single-cell eQTL mapping of 393 previously genotyped segregants.

**Figure supplement 2.** Cell-cycle classification of the single cells from the set of 393 previously genotyped segregants visualized on different combinations of principal components.

**Figure supplement 3.** Marker gene expression of cell-cycle classified single cells from the set of 393 previously genotyped segregants.

**Figure supplement 4.** Cell-cycle classification of the single cells from the set of 393 previously genotyped segregants.

**Figure supplement 5.** Cell-cycle marker gene expression of the single cells from the set of 393 previously genotyped segregants.

**Figure supplement 6.** Histogram of the number of single cells identified for each of the 393 segregants.

**Figure supplement 7.** Single-cell hidden Markov model (HMM) genotyping accuracy compared to the number of unique molecular identifiers (UMIs) per cell.

**Figure supplement 8.** Local eQTL effects estimated with different genotyping methods.

each cell to one of the 393 segregants. We observed a median of 17 cells per segregant, with 277 of the segregants sampled more than ten times (**Figure 1—figure supplement 6**). The genotypes measured from scRNA-seq data were in high agreement with those obtained from whole-genome sequencing of the same strains (median genotype agreement 92.5%). The agreement was higher in cells with more UMIs (**Figure 1—figure supplement 7**), and we leveraged higher yields of UMIs per cell in subsequent experiments to ensure better genotyping accuracy.

We used the two sets of genotypes to map local eQTLs—those that influence the expression of nearby genes, most commonly in cis. We modeled the genetic effects of the closest marker to each transcript on single-cell gene expression with a count-based model that did not include a cell-cycle term. We mapped 770 local eQTLs at an FDR of 5% with the HMM-based genotypes, and 697 with the matched genotypes obtained from whole-genome sequencing of the segregants; 611 eQTLs were detected in both analyses (**Supplementary file 1, table S6**). We further compared the local eQTL effects for all 4901 tested transcripts, regardless of statistical significance, and found that they were highly correlated between the two sets of genotypes (Spearman's ρ = 0.93, p < 10$^{-15}$; **Figure 1— figure supplement 8**). A single-cell eQTL study on a different set of previously genotyped segregants from the same cross reached a similar conclusion (**N'Guessan et al., 2023**), providing further evidence that genotypes obtained from scRNA-seq data at transcribed SNPs are of sufficient quality for eQTL mapping.

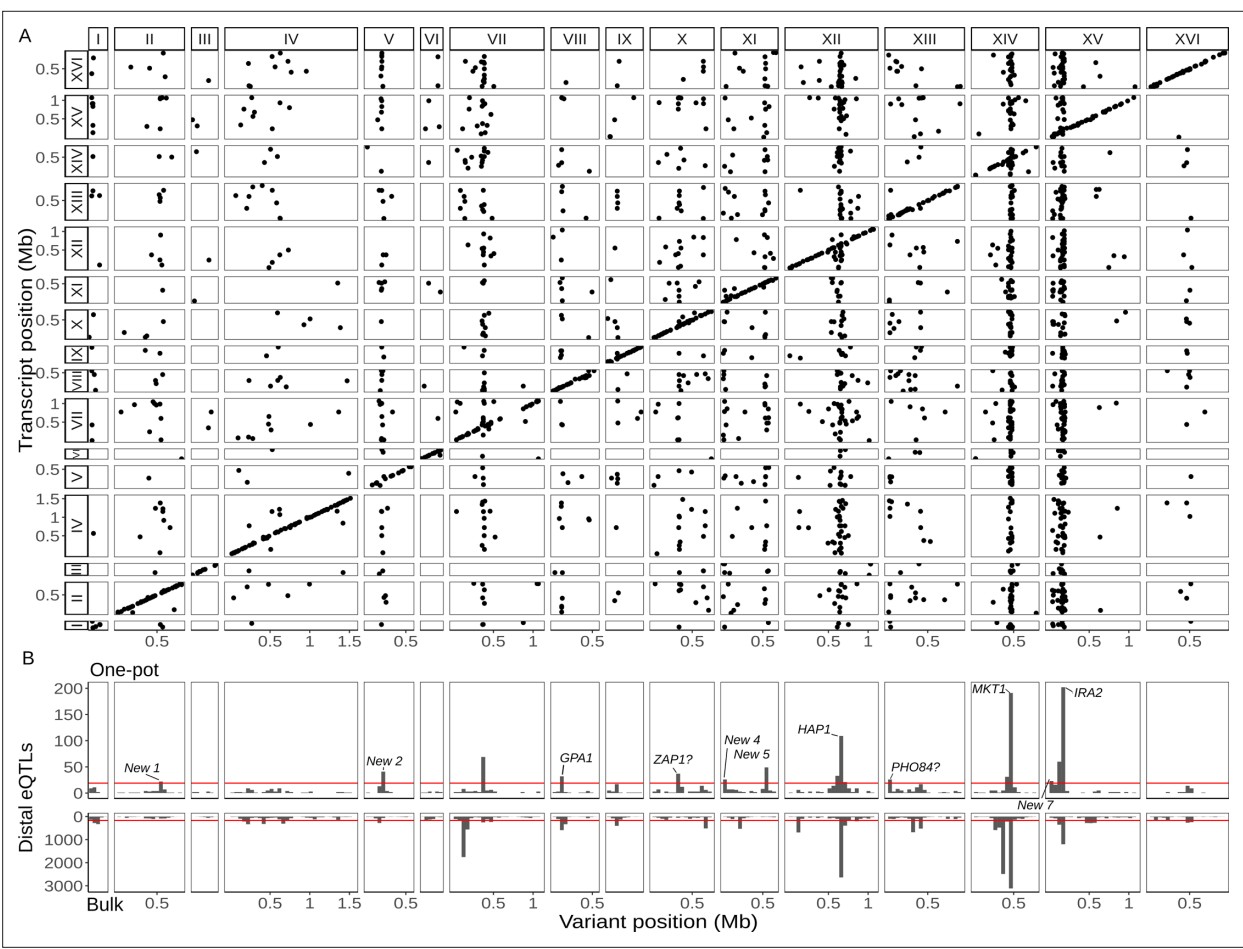

**Figure 2.** Single-cell eQTL map recapitulates bulk *trans*-eQTL hotspots and identifies new hotspots. (**A**) Map of local and distant eQTLs. Each point denotes an eQTL, with the genomic position of the peak marker on the *x*-axis and the genomic location of the gene with the expression difference on the *y*-axis. The high density of points on the diagonal line with a slope of one indicates that many genes have local eQTLs. The dense vertical bands correspond to *trans*-eQTL hotspots. (**B**) Histogram showing the number of distant eQTLs in 50 kb windows top: one-pot eQTL map; bottom: bulk eQTL map (**Albert et al., 2018**). Red lines show statistical eQTL enrichment thresholds for a window to be designated a hotspot. Text labels highlight known and putative causal genes underlying hotspots, as well as loci that meet hotspot criteria only in the current study.

## One-pot eQTL mapping in de novo yeast segregants

One-pot eQTL mapping is an attractive experimental design compared to bulk RNA sequencing and genotyping because it lowers cost, eliminates individual sample preparation, and reduces other sources of technical variation. To compare one-pot eQTL mapping with the traditional bulk design, we generated segregants de novo from a cross between BY and RM (*Albert et al., 2018*). We used scRNA-seq to measure the expression of 5435 transcripts in 27,744 single cells (*Supplementary file 1, table S4*). We mapped 1031 local eQTLs at an FDR of 5%. We compared these results to those from bulk RNA-seq and genotyping in the same cross (*Albert et al., 2018*) and found that 717 (69.5%) of the 1031 local eQTLs were also detected as statistically significant in that study, with an additional 108 local eQTLs showing effects in the same direction (*Figures 1C and 2A*; *Supplementary file 2, table S1*). Thus, 825 (80%) of the local eQTLs detected with the one-pot approach were supported by the bulk results, despite differences in growth conditions and experimental procedures between the two studies.

We next broadened our analysis to the *trans*-acting (distant) eQTLs, here defined as those that influence the expression of genes on a different chromosome. We mapped 1562 distant eQTLs at an FDR of 5% (*Figure 2A*; *Supplementary file 3, table S1*). As in previous studies (*Albert et al., 2018*; *Brem et al., 2002*), distant eQTLs were not uniformly distributed throughout the genome, but rather clustered at a number of hotspot loci that influence the expression of many genes. We identified 12 distant eQTL hotspots in the one-pot eQTL experiment (*Supplementary file 3, table S1*). When we applied the same criteria for defining a hotspot, we identified 21 hotspots in the bulk eQTL experiment in the same cross (*Figure 2B*). Five regions met the hotspot criteria in both studies, including the well-characterized hotspots driven by variants in the genes *MKT1* (*Zhu et al., 2008*), *GPA1* (*Yvert et al., 2003*), *IRA2* (*Smith and Kruglyak, 2008*), and *HAP1* (*Brem et al., 2002*). One hotspot on chromosome XIV in the bulk experiment was not observed here because it is caused by a de novo variant in the gene *KRE33* that arose in the RM parent used in the construction of the bulk eQTL mapping panel (*Albert et al., 2018*; *Jerison et al., 2017*). The other hotspots from the bulk experiment generally affected the expression of fewer genes, and the fact that they did not meet hotspot criteria here can be explained by a combination of statistical power and different experimental conditions.

To learn more about the seven regions that met hotspot criteria only in the single-cell experiment but not in the bulk experiment, we performed a functional enrichment analysis of the genes they influence (*Figure 2B*; *Supplementary file 3, table S1*). The hotspot on chromosome X at position 323,158 changed the expression of 36 genes that were enriched for gene ontology (GO) terms related to zinc ion transmembrane transporter activity and transition metal ion transmembrane transporter activity. The hotspot region contains the gene *ZAP1*, which encodes a zinc-regulated transcription factor (*Zhao and Eide, 1997*). This gene contains nine missense variants between BY and RM, and we predict that *ZAP1* is the causal gene underlying this hotspot (see also *Weith et al., 2023*). The hotspot on chromosome XIII at position 24,326 changed the expression of 26 genes that were enriched for GO terms related to acid phosphatase activity. The hotspot region contains the gene *PHO84*, which encodes an inorganic phosphate transporter (*Bun-Ya et al., 1991*). BY harbors a rare coding variant P259L in *PHO84* (L allele frequency = 0.3%) (*Peter et al., 2018*) that has been shown to affect resistance to polychlorinated phenols (*Perlstein et al., 2007*) and is the likely causal variant for this hotspot. The other five hotspots were enriched for GO terms broadly related to growth. We grew the yeast segregants for scRNA-seq in a medium containing sheath fluid, a phosphate-buffered saline (PBS) solution with a pH of 7.4, whereas unbuffered minimal medium was used in the bulk eQTL study. Gene–environment interactions in gene expression are common in yeast (*Smith and Kruglyak, 2008*), especially for distant eQTLs, and subtle differences in the growth conditions between the two studies may explain why these new loci met the hotspot criteria only in the single-cell study.

We took advantage of the convenience of one-pot eQTL mapping and applied it to two additional yeast crosses, one between a clinical strain (YJM145) and a soil strain (YPS163) both isolated in the United States (44,784 cells, 5556 transcripts; *Supplementary file 1, table S4*), and another between a soil strain isolated in South Africa (CBS2888) and a clinical strain isolated in Italy (YJM981) (6595 cells, 4696 transcripts; *Supplementary file 1, table S4*). Hereafter, we refer to the BY × RM cross as cross A and the two new crosses as crosses B and C, respectively. We mapped a total of 1914 local eQTLs in the new crosses (1193 in cross B and 721 in cross C; *Supplementary file 2, tables S2 and S3*), as well as 1626 distant eQTLs (550 in cross B and 1126 in cross C; *Figure 3A, B*;

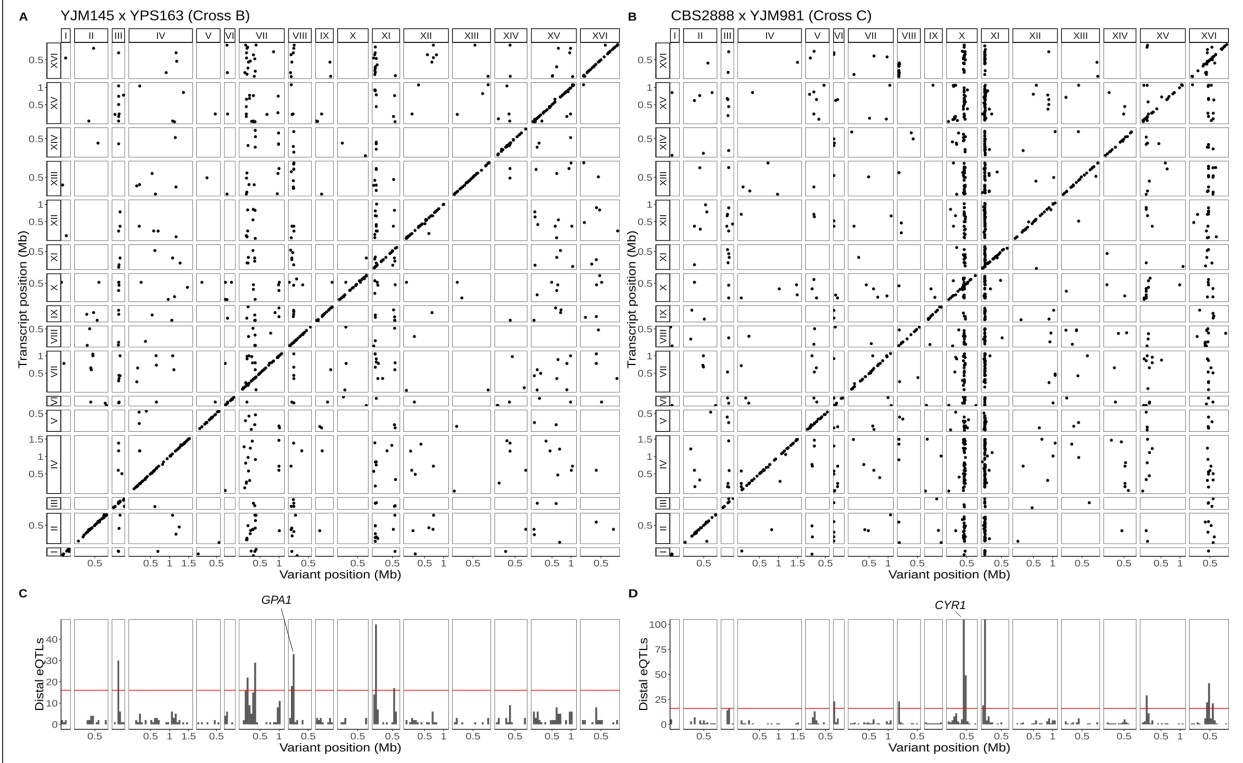

**Figure 3.** Single-cell eQTL maps in two new crosses. (**A**) eQTL map for the YJM145 × YPS163 cross (cross B). (**B**) eQTL map for the CBS2888 × YJM981 cross (cross C). (**C**) Histogram of distal eQTLs showing hotspots in cross B. (**D**) Histogram of distal eQTLs showing hotspots in cross C. The y-axis has been truncated to have a maximum value 100 for ease of visualization purposes. The hotspot on chromosome X near the gene *CYR1* influences the expression of 175 genes, and the hotspot on chromosome XI influences the expression of 386 genes.

*Supplementary file 3, tables S2 and S3*). These distant eQTLs clustered into 13 hotspots (6 in cross B and 7 in cross C; *Figure 3C, D*). Of the 25 hotspots detected in the three crosses, 14 (56%) were unique to a single cross (7/12 in cross A, 2/6 in cross B, and 5/7 in cross C). This observation is consistent with prior work suggesting that variants with widespread effects on gene expression are likely to be deleterious, and that purifying selection should reduce their allele frequencies, making them more likely to be strain specific (*Ronald and Akey, 2007*). We used functional annotations to identify candidate genes for two of the new hotspots: *GPA1* for the hotspot on chromosome VIII in cross B (*Supplementary file 4*—Cross B chrVIII:46887–140660) and *CYR1* for the hotspot on chromosome X in cross C (*Supplementary file 4*—Cross C chrX:397734–497167); the biological effects of these hotspots are discussed below.

## Distant eQTLs effects are more dependent on cell-cycle stage than local eQTLs effects

We asked whether the effects of eQTLs varied across the different stages of the cell cycle. Of the 2945 total local eQTLs detected in the three crosses, only 116 (4%) showed significant interactions between the eQTL effect and the cell-cycle stage at an FDR of 5%. In contrast, 790 (24.4%) of 3238 distant eQTLs showed significant interactions with the cell-cycle stage (OR = 7.8, Fisher's exact test, $p < 10^{-15}$), which suggests that the effects of distant eQTLs depend more on the state of a cell than those of local eQTLs (*Supplementary files 2 and 3*). This observation is consistent with prior work which showed that the effects of distant eQTLs are often dependent on the environment (*Smith and Kruglyak, 2008*), tissue (*Aguet et al., 2020*; *Battle et al., 2017*), and cell type (*Ben-David et al., 2021*), while those of local eQTLs tend to be less affected by these factors, perhaps because their effects on expression are more direct. Our results extend this notion beyond external environments, tissues and cell types to internal cellular states in a single-cell type.

## Identification of hundreds of genetic effects on expression noise

An outstanding question in genetics is whether, and to what extent, genetic variation influences noise in gene expression—that is, do some genetic variants alter the variability in the expression level of specific genes, separately from their effects on the average expression levels? Measurement of expression in single cells with different genotypes is uniquely suited to exploring this question, but separating the effects on noise from those on average expression is not trivial, and previously identified genetic effects on expression noise in scRNA-seq data could be explained by their effects on average expression (*Sarkar et al., 2019*). In mapping panels, apparent allelic effects on intrinsic expression variability can instead reflect extrinsic sources of expression variability that differ between cells, such as cell-cycle stage and genetic differences in *trans*-acting factors. To overcome these issues of interpretation, we investigated the genetic contribution to intrinsic noise in gene expression in scRNA-seq data we generated for F1 diploid hybrids of the parental strains. The F1 diploid yeast cells are isogenic and share all *trans*-acting factors, allowing us to exclude extrinsic genetic sources of expression variability and focus on allele-specific contributions to gene expression noise.

We obtained a total of 13,973 single-cell transcriptomes from F1 diploids used to generate the segregants for the three crosses (5890 for cross A, 2864 for cross B, and 5219 for cross C; *Supplementary file 1, table S4*). We classified each cell into one of four cell-cycle stages relevant for diploids (M/G1, G1/S, S, and G2/M). We found 3406 genes with allele-specific effects on average expression levels (668 for cross A, 996 for cross B, and 1742 for cross C; *Supplementary file 5*). These allele-specific effects were well correlated with local eQTL effects from the eQTL mapping experiments described above (*Figure 4—figure supplement 1*). We observed 160 genes with significant interactions between allele-specific expression and cell-cycle stage (*Supplementary file 5*).

We next looked for allele-specific effects on gene expression noise. We used an approach that tests for significant differences in gene expression noise between the two alleles in the F1 diploid hybrids after accounting for average differences in gene expression due to genotype, cell-cycle stage, and their interactions (*Figure 4A, B*, *Figure 4—figure supplement 2*; Methods). Using this approach, we found a total of 874 genes with allele-specific effects on expression noise at an FDR of 5%, independent of any effects on average expression (*Figure 4C*; *Supplementary file 6*).

An additional consideration for these analyses is that prior work has revealed an empirical negative correlation between gene expression noise and average gene expression, even when noise is estimated while accounting for average expression (*Antolović et al., 2017*; *Love et al., 2014*). We observed this global trend in our data—across all genes, noise was negatively correlated with expression level (Spearman's $\rho = -0.42$, $p < 10^{-15}$; *Figure 4—figure supplement 3*). To test whether the observed allele-specific effects on expression noise could arise from this trend, we asked whether the confidence interval (CI) of each significant allele-specific effect on noise overlapped the CI of the global trend line (Methods). Of the 874 genes with allele-specific effects on noise, these CIs did not overlap for 377, suggesting that these allele-specific effects on noise cannot be explained by the empirical global relationship between expression noise and average expression (*Figure 4C*; *Supplementary file 6*).

An illustrative example of an allele-specific effect on expression noise was found in crosses A and C for the gene *HSP12*, which encodes an intrinsically unstructured protein that improves membrane stability (*Supplementary file 6*; *Welker et al., 2010*). *HSP12* is a member of the general stress response pathway regulated by Msn2/4 (*Kuang et al., 2017*) and helps yeast cells survive high-temperature shocks (*Welker et al., 2010*). In cross A, the RM allele did not significantly change the expression of *HSP12* compared to the BY allele, but it did significantly increase the noise, whereas in cross C, the YJM981 allele decreased the expression of *HSP12* compared to the CBS2888 allele and increased the noise. Previous work found that *HSP12* has high extrinsic expression noise relative to other genes (*Stewart-Ornstein et al., 2012)*, and it was proposed that the high noise arises from variability in the activity of the Msn2/4 stress response pathway and subsequent activation of Msn2/4 targets such as *HSP12* (*Gasch et al., 2017*). Our experiments in F1 hybrids control for sources of extrinsic noise, such as Msn2/4 activity, and our results suggest that the RM and YJM981 alleles of *HSP12* are intrinsically more variable than the BY and CBS2888 alleles, and that genetic differences acting in cis are responsible. We hypothesize that the RM and YJM981 allele of *HSP12* may provide a

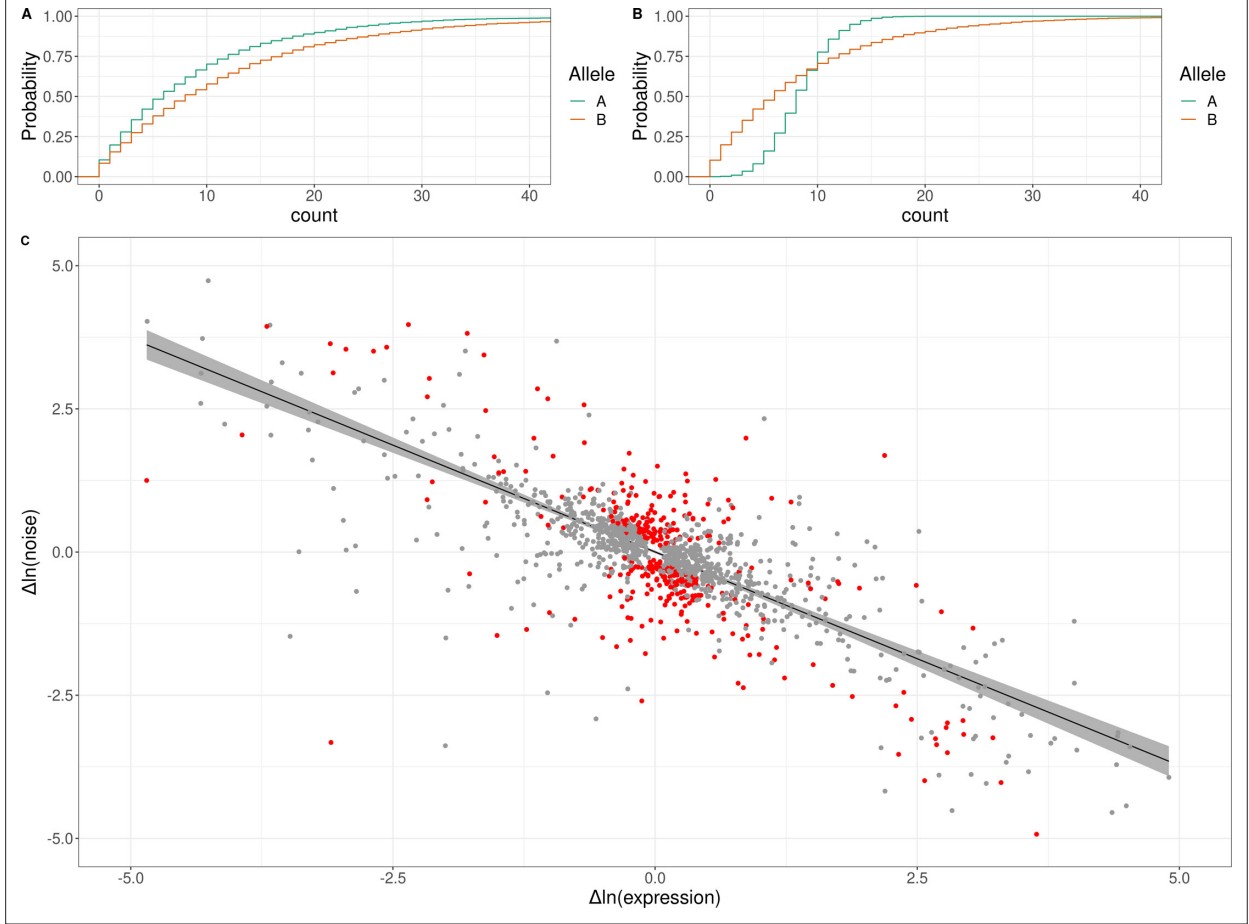

**Figure 4.** Genetic effects on expression noise. (**A**) Cumulative distribution of simulated allele-specific counts for two alleles with different average expression but the same expression noise. (**B**) Cumulative distribution of simulated allele-specific counts for two alleles with different expression noise but the same average expression. These simulated distributions are shown to illustrate allele-specific effects on average expression and on expression noise, respectively. (**C**) Log–log scatter plot of change in expression noise between alleles (*x*-axis) against change in average expression between alleles (*y*-axis); points correspond to all 1487 genes with significant allele-specific effects on expression noise and/or average expression. Black line shows the predicted change in noise given a change in expression, with the 95% confidence interval for the trend shown in gray. The 377 genes with allele-specific effects on expression noise that cannot be accounted for by the overall trend are shown in red. The *x* and *y* axes have been truncated at –5 and 5 for ease of visualization purposes, which left out 30 of 1487 data points.

The online version of this article includes the following figure supplement(s) for figure 4:

**Figure supplement 1.** The allele-specific expression effects compared to the local eQTL effects from each cross.

**Figure supplement 2.** Estimates of noise are biased downwards at low expression levels.

**Figure supplement 3.** Empirical global relationship between expression noise and average expression levels.

fitness advantage during periods of extreme stress via a bet-hedging strategy in which noisy expression of *HSP12* creates a subpopulation of cells with very high *HSP12* expression that can better survive environmental shocks.

## Natural genetic variants affect cell-cycle occupancy

Because the single-cell expression data allowed us to assign each genotyped cell to a stage of the cell cycle, we next moved beyond gene expression and searched for genetic effects on cell-cycle progression. Specifically, we looked for loci at which one allele is overrepresented in cells assigned to a particular cell-cycle stage. Because cell-cycle occupancy represents the proportion of cells assigned to a given stage, changes in the proportion of cells in each stage are correlated, potentially leading to QTLs with effects on occupancy of multiple cell-cycle stages. We found a total of 20 unique cell-cycle occupancy QTLs in the three crosses (4 for cross A, 10 for cross B, and 6 for cross C; *Figure 5A*;

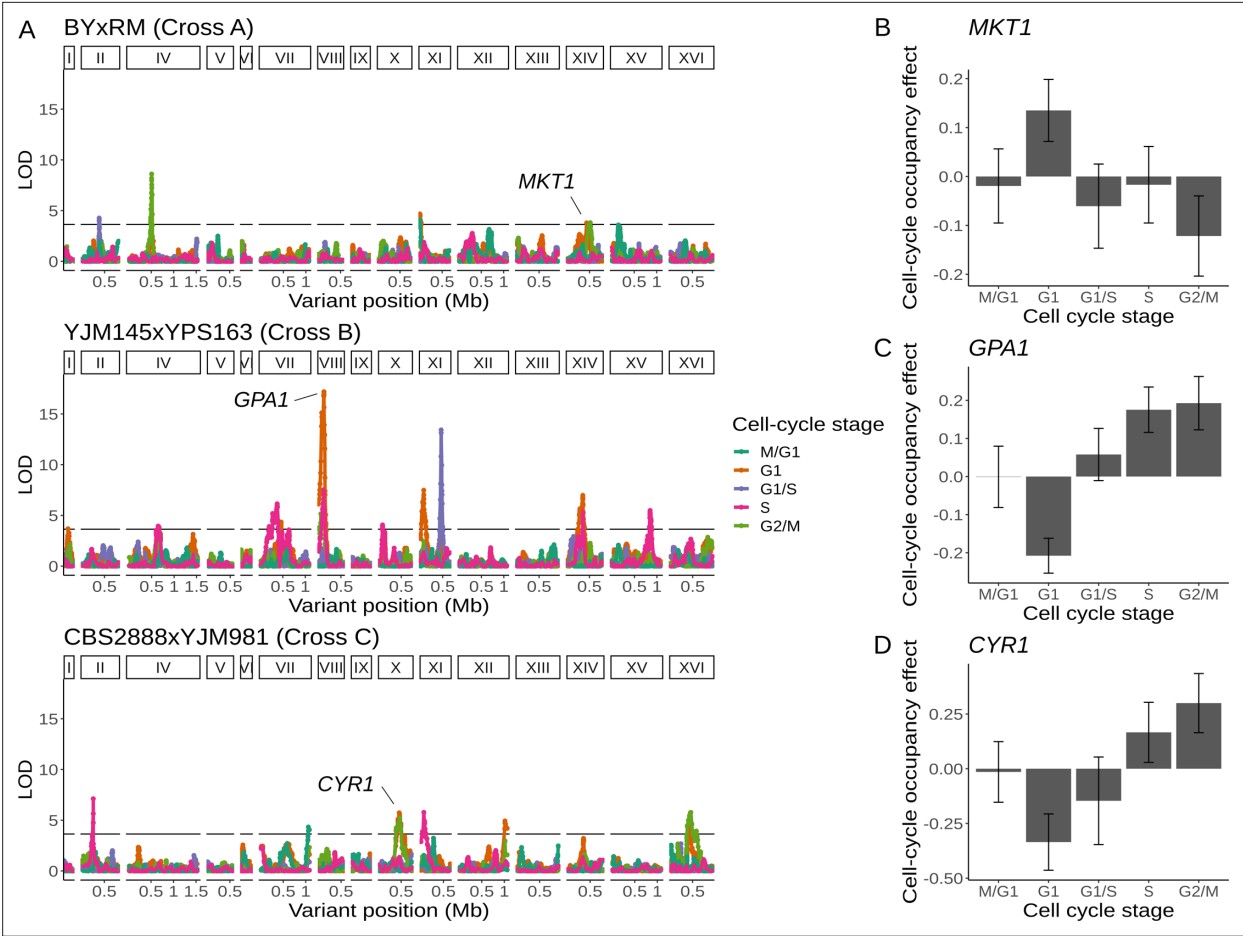

**Figure 5.** Natural genetic variants affect cell-cycle occupancy. (**A**) Cell-cycle occupancy QTL map for three different crosses. LOD score for linkage with cell-cycle occupancy (*y*-axis) is plotted against the genomic location of genetic markers (*x*-axis). Colored lines show results for different cell-cycle stages as denoted in the legend. Horizontal line corresponds to a family-wise error rate (FWER) threshold of 0.05. Text labels highlight genes with QTL effects shown in panels B–D. Cell-cycle occupancy mapping was not performed on chromosome III. (**B**) Variation in *MKT1* increases G1 occupancy and decreases G2/M occupancy in the BY × RM cross. (**C**) Variation in *GPA1* decreases G1 occupancy and increases S and G2/M occupancy in the YJM145 × YPS163 cross. (**D**) Variation in *CYR1* decreases G1 occupancy and increases S and G2/M occupancy in the CBS2888 × YJM981 cross. Error bars in B–D represent 95% confidence intervals.

The online version of this article includes the following figure supplement(s) for figure 5:

**Figure supplement 1.** Single-cell expression profiling of allele-replacement strains comparing the cell-cycle distribution of strains with the 82R allele of *GPA1* to strains with the WT allele of *GPA1*.

*Supplementary file 7*). One of the QTLs identified in cross A contained the gene *MKT1*, variation in which is known to affect dozens of growth traits and thousands of molecular traits, including gene expression (*Figure 5B*; *Zhu et al., 2008*). Segregants inheriting the RM allele of *MKT1* are overrepresented in the G1 stage of the cell cycle and underrepresented in the G2/M stage. This observation suggests that some of the previously described cellular impacts of *MKT1* variation may arise as a consequence of its effect on progression of yeast cells through the cell cycle.

We identified a cell-cycle occupancy QTL on chromosome X in cross C whose location coincided with a distant eQTL hotspot (*Figures 3D and 5D*). This hotspot affected the expression of 224 genes that were enriched for GO terms related to oxidative phosphorylation and the citric acid cycle. We combined the eQTL mapping results with growth QTL (*Bloom et al., 2019a*) from the same cross and predicted that the likely causal gene underlying this hotspot is *CYR1* (*Supplementary file 4*—Cross C chrX:397734–497167). *CYR1* encodes adenylate cyclase, an enzyme which catalyzes the reaction that produces cyclic AMP (*Matsumoto et al., 1982*). Segregants which carry the *CBS2888* allele of *CYR1* more frequently occupy the G1 phase of the cell cycle and show improved growth in eight stressful

conditions (*Figure 5D*). The *CBS2888* allele of *CYR1* contains multiple variants with predicted large deleterious effects on the gene, and these variants may act individually or together to compromise the function of *CYR1*, with the result that cells with this natural allele may mimic the G1 arrest phenotype observed in temperature-sensitive mutants of *CYR1* (*Matsumoto et al., 1983*). Mutations in *CYR1* are known to alter stress tolerance in yeast (*Versele et al., 2004*; *Vanhalewyn et al., 1999*; *Vianna et al., 2010*), providing additional support for our hypothesis that *CYR1* is the causal gene underlying this hotspot.

## The W82R variant of *GPA1* alters gene expression and cell-cycle occupancy

We mapped a cell-cycle occupancy QTL in cross B to a region on chromosome VIII that contains the gene *GPA1* (*Figure 5C*). *GPA1* encodes the GTP-binding alpha subunit of a heterotrimeric G protein that mediates the response to mating pheromone (*Nakafuku et al., 1987*). Segregants carrying the YJM145 allele of *GPA1* more frequently occupied G1, the cell-cycle stage during which the mating pathway is active (*Figure 5A*; *Lang et al., 2009*). This locus is also a distant eQTL hotspot that influenced the expression of 51 genes (*Figure 2C*). These genes are enriched for GO terms related to sexual reproduction and cellular response to pheromone (*Supplementary file 4*—Cross B chrVIII:46887–140660). Variants in the coding sequence of *GPA1* are known to alter the expression of genes involved in mating (*Yvert et al., 2003*). The YJM145 allele of *GPA1* contains a variant that changes a tryptophan to an arginine at position 82 of the Gpa1 protein. An evolutionary analysis revealed that this residue has been conserved as tryptophan for ~400 million years in the budding yeasts (*Saccharomycotina*) (*Shen et al., 2018*), and is commonly found as a aromatic amino acid (phenylalanine, tyrosine, or tryptophan) across the tree of life (*Supplementary file 8*). The conservation of this tryptophan is reflected in the prediction that the 82R allele is highly deleterious to the function of Gpa1 (Provean score of −13.915) (*Choi and Chan, 2015*). We thus hypothesized that this variant in *GPA1* is responsible for the observed effects of the chromosome VIII locus on both gene expression and cell-cycle occupancy.

To test this hypothesis, we used CRISPR–Cas9 to engineer each allele of the W82R variant into a common genetic background (*Supplementary file 1, table S1*; *Sadhu et al., 2018*). We performed scRNA-seq on 26,859 cells from these isogenic strains that differed only in whether they carried the 82R ($N$ = 11,695) or the 82W allele ($N$ = 15,164) of *GPA1*. We observed that for 36 of the 50 genes affected by the hotspot and detected in our single-cell validation dataset, the sign of the expression difference was consistent between the eQTL effect and the W82R validation experiment (binomial test, p = 0.0026; *Supplementary file 9*). Importantly, the gene expression difference in the W82R experiment was statistically significant and concordant with the eQTL effect for all six mating-related genes affected by this hotspot (*AGA1*, *AGA2*, *MFA1*, *STE2*, *FUS3*, and *PRM5*), showing that the 82R allele isolated from other segregating genetic variation increases the expression of genes involved in the mating response. Consistent with the QTL effect, cells with the 82R allele were overrepresented in G1 (46.2% in G1 vs. 42.8% in other stages, logistic regression, p < $10^{-15}$; *Figure 5—figure supplement 1*). We conclude that the W82R variant is responsible for the observed effects of this chromosome VIII QTL on gene expression and cell-cycle occupancy.

## The 82R allele of *GPA1* increases mating efficiency at the cost of growth rate

One possible consequence of increasing the proportion of cells in G1 is slowing progression through the cell cycle, with a corresponding decrease in the cell doubling rate. Previous work has shown that strains which carry a different variant in *GPA1* (a G to T substitution at position 1406 in the coding sequence of the gene, which results in a serine to isoleucine substitution at position 469 of the protein) have decreased growth rates (*Lang et al., 2009*). We measured growth rates of the engineered strains and found that cells with the 82R allele grew slower than those with the 82W allele (relative growth rate = 0.993, $T$ = −2.592, p = 0.0268), but that this effect was smaller than that observed for the S469I variant (relative growth rate of the 469I allele = 0.984, $T$ = −5.291, p < 0.001; *Figure 6A*). The 469I allele is known to improve the efficiency of mating, a difference we successfully replicated (relative 469I mating efficiency = 110%, $T$ = 9.73, p < 0.001). We observed that the 82R allele also increased mating efficiency (relative mating efficiency = 107%, $T$ = 6.75, p < 0.001), but to a lesser extent than

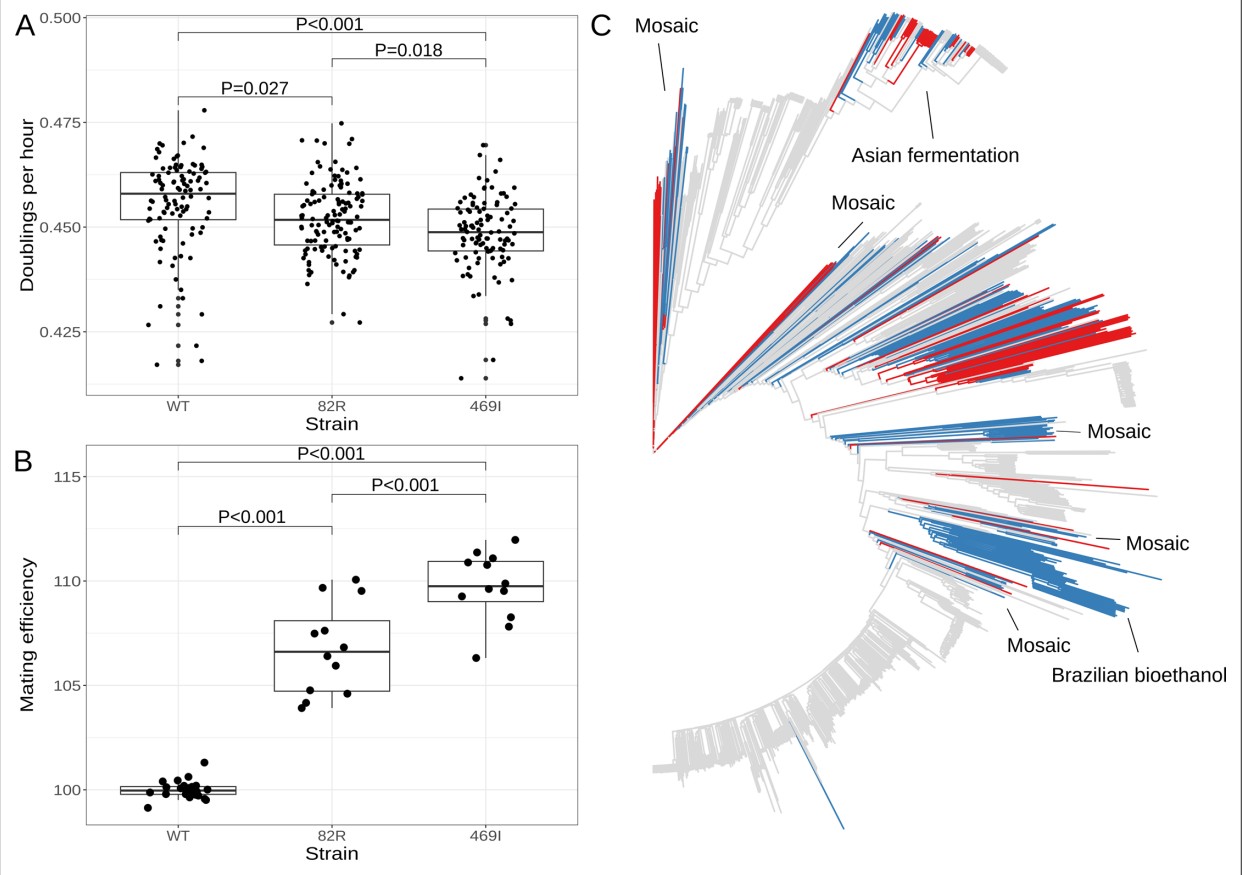

**Figure 6.** The 82R allele of *GPA1* increases mating efficiency at the cost of growth rate and is associated with increased outbreeding in natural populations. (**A**) Boxplots show growth of allele replacement strains grown in glucose. Points represent replicate measurements of the doublings per hour for each strain. Tukey's HSD adjusted p-values of pairwise comparisons of allele replacement strains are shown. (**B**) Boxplots show mating efficiency of allele replacement strains; details as in **A**. (**C**) Genome-wide neighbor-joining tree of 1011 sequenced yeast isolates. Strains in which only the 82R allele is present are denoted in blue; strains with support for both 82R and 82W alleles are denoted in red; and strains in which only 82W allele is present are denoted in gray. We observed that the 82R allele is enriched in mosaic strains (allele frequency = 45.3%, permutation test p = 0.007). Other clades mentioned in the text are labeled on the tree.

The online version of this article includes the following figure supplement(s) for figure 6:

**Figure supplement 1.** Mating efficiency of the 82R allele of *GPA1* compared to the 469I and wild-type (82W 469S) alleles.

**Figure supplement 2.** Allele frequency of the 82R allele of *GPA1* in sequenced yeast strains.

**Figure supplement 3.** Genome-wide heterozygosity per strain, after classifying strains as having the 82W allele or 82R allele of *GPA1*.

the 469I allele (82R mating efficiency compared to 469I = 97.2%, *T* = −2.98, p < 0.001; *Figure 6B*, *Figure 6—figure supplement 1*). These results show that natural variants in *GPA1* increase mating efficiency at the cost of slower growth. Both effects may be explained by the impact of these variants on the mating pathway—enhanced activity of the pathway facilitates mating in the presence of partners, while inappropriate pathway activation in the absence of partners slows down the G1 phase of the cell cycle, as we have shown for the 82R allele, thereby decreasing growth rate.

## The W82R allele of *GPA1* is common in the yeast population and is associated with increased outbreeding in natural populations

We searched for the 82R and 469I alleles in a worldwide collection of 1011 *S. cerevisiae* isolates (*Peter et al., 2018*) and found that the 469I allele is rare in the population (1.9%), whereas the 82R allele is common (20.5%) (*Figure 6C*). The 82R allele is fixed in a clade of strains isolated from Brazilian bioethanol (82R allele frequency = 100%) and is found at high frequency (58%) in a clade of strains isolated from Asian fermentation products such as rice wine (*Figure 6—figure supplement 2*). Peter et al.

identified four groups of mosaic strains, which are characterized by admixture of two or more different lineages through outbreeding, and we observed that the 82R allele is enriched in these mosaic strains (allele frequency = 45.3%, permutation test p = 0.007). Strains derived from outbreeding events between genetically distinct parents are expected to show higher rates of heterozygosity than strains resulting from inbreeding or clonal propagation. We compared homozygous (<5% heterozygous sites) and heterozygous (>5% heterozygous sites) strains, as defined by Peter et al., and found that the 82R allele is enriched in heterozygous strains (OR = 3.3, Fisher's exact test, $p < 10^{-7}$) and associated with higher rates of heterozygosity (Wilcoxon rank sum test, $p < 10^{-15}$; *Figure 6—figure supplement 3*). We have shown that the 82R allele increases mating efficiency in the lab, and these observations suggest that this increased mating efficiency may translate into higher outcrossing rates in nature.

## Discussion

We used a one-pot single-cell eQTL mapping design, in which tens of thousands of cells from a segregating population are subjected to scRNA-seq, to map thousands of eQTLs in three different yeast crosses. We identified both local and distant eQTLs and showed that distant eQTLs in all three crosses cluster at hotspot loci that affect the expression of many genes, recapitulating and extending previous observations made with bulk eQTL mapping in the widely studied BY–RM cross. Notably, most of these hotspots are not shared between crosses, which suggests that they are caused by alleles unique to one of the six parent strains. This observation is consistent with the idea that alleles that alter the expression of many genes are likely to be selectively disfavored and therefore present at lower frequencies in the yeast population (*Ronald and Akey, 2007*; *Bloom et al., 2019a*).

Prior research has leveraged scRNA-seq data to detect genetic loci that alter gene expression noise, but their effects could not be separated from those on average expression levels and may reflect other sources of extrinsic cell-to-cell variability (*Sarkar et al., 2019*). To account for extrinsic factors, we obtained thousands of transcriptomes from single cells of hybrid diploid F1 yeast and tested for allele-specific differences in intrinsic gene expression noise. We employed an approach that accounts for average changes in gene expression and identified 874 genes with an allele-specific effect on gene expression noise. For 377 of these genes, the effects on noise could not be accounted for by the empirically observed negative correlation across genes between estimated gene expression noise and average gene expression (*Love et al., 2014*). We observed allele-specific effects on *HSP12* expression noise in two separate crosses. *HSP12* plays a role in protection against high-temperature shocks, and the high-noise alleles may provide a fitness advantage during high-temperature stress by creating a subpopulation of cells with very high *HSP12* expression that can survive under these conditions. This observation adds to previous reports showing that noise mediated by promoter variants can provide a fitness advantage in times of environmental stress (*Liu et al., 2015*) and may constrain variation in promoter evolution (*Metzger et al., 2015*).

Single-cell RNA-seq data allowed us to assign each cell to a cell-cycle stage and explore genetic effects on expression during different stages of the cell cycle. We detected hundreds of eQTLs whose effects differed across cell-cycle stages. Distant eQTLs were more likely than local eQTLs to be cell cycle dependent, perhaps because the effects of distant eQTLs are more indirect and mediated by cellular regulatory networks that are affected by the cell cycle (*Albert et al., 2018*). Previous work has shown that effects of distant eQTLs are more sensitive than those of local eQTLs to tissue type (*Battle et al., 2017*) and external environment (*Smith and Kruglyak, 2008*), and our results extend these findings to show that they are more sensitive to internal cellular states within a single-cell type.

We used the ability to classify genotyped cells by their cell-cycle stage to identify 20 loci that altered the occupancy of different cell-cycle stages, one of which overlapped an eQTL hotspot. We used fine mapping and allele replacement with CRISPR–Cas9 to show that a common variant (W82R) in the gene *GPA1* is responsible for the effects of this locus on cell-cycle occupancy and gene expression. We further showed that the 82R allele increases yeast mating efficiency at the cost of slower growth. Natural yeast isolates vary in their propensity to mate or enter the cell cycle upon germination (*McClure et al., 2018*), leading us to ask whether the 82R allele alters mating efficiency outside the lab. We searched for this allele in a collection of 1011 sequenced yeast isolates (*Peter et al., 2018*) and found that it is common (20.5%) and occurs more frequently in isolates that show evidence of recent outcrossing, suggesting that the observed increase in mating efficiency in the lab translates into more frequent mating in nature. Outcrossing rate has a major impact on the genetic structure of

a population and its response to natural selection (*Hartfield et al., 2017*), and our results suggest that common variants can alter this key evolutionary parameter.

Studies of genetic effects on gene expression provide a molecular lens into the genetic basis of complex traits. One-pot single-cell eQTL mapping makes such studies cheaper, more efficient, and more flexible. This approach will power broader explorations of how genetic variants influence gene expression in different genetic backgrounds and under different experimental conditions. It also enables integration of information across multiple levels, as shown here for the case of gene expression, cell-cycle occupancy, and mating efficiency. The results of this study have the potential to inform the design, execution, and analysis of other one-pot studies of the effects of genetic variation on gene expression, such as human 'cell villages' (*Wells et al., 2023*; *Neavin et al., 2023*).

# Materials and methods

Unless otherwise specified, computational analyses were performed in the R (v4.4.0) programming language (*R Development Core Team, 2022*) and visualizations were created using the ggplot2 package (v3.5.1) (*Wickham, 2009*).

Strains, plasmids, and primers used in this study are listed in *Supplementary file 1, tables S1-S3*.

## Sporulation and single-cell sorting

To assess our ability to reconstruct genotypes and perform eQTL mapping with single-cell data, we performed a single-cell eQTL study using 480 MATa segregants from a cross between a lab (BY) and wine (RM) strain (YLK1993, BY × RM) that were genotyped by whole-genome sequencing and had their transcriptomes measured by bulk RNA-seq (*Albert et al., 2018*). These 480 segregants were grown to saturation in 96-well plates and pooled at equal volumes to create our study population. This pool of strains was then transferred to minimal YNB medium (6.7 g/l Difco Yeast Nitrogen Base w/o Amino Acids, 2% glucose) and grown overnight at 30°C in YPD (2% bacto peptone, 1% yeast extract, 2% glucose). This pool of segregants was diluted to an OD600 of ~0.05 and allowed to grow until they were in mid-log, defined as the culture having an OD600 of between 0.4 and 0.6. Cells were then harvested using a 125-ml vacumm filtration system (Sigma-Aldrich #Z290467) fitted with a 0.2-μM nylon membrane filters (Sterlitech, #NY0225100). The filter was transferred to a 50-ml conical tube and submerged in liquid nitrogen to flash freeze the yeast cells. The conical tubes were transferred to a –80°C freezer for later use. 393 of the 480 (81.9%) MATa segregants were present in the scRNA-seq data.

For de novo eQTL mapping, we used three parental diploids that were transformed with a fluorescent Magic marker that we have previously described (*Treusch et al., 2015*). These crosses were YLK3051 (BY × RM; cross A; PLK124), YLK3301 (YJM145 × YPS163; cross B; PLK124), YLK3004 (CBS2888 × YJM981; cross C; PLK73; Tables S3 and S4). Spores containing haploid recombinant progeny were obtained from these diploid strains by growing the parental diploid strains for 5–7 days in SPO++ sporulation medium at room temperature (https://dunham.gs.washington.edu/sporulationdissection.htm). We used a modified random spore prep protocol to obtain individual spores prior to fluorescence-activated cell sorting (FACS). We resuspended 1 ml of the sporulation cultures in 100 µl of water and added 20 µl of 2000 U/ml Zymolyase 20T (Amsbio, #120491-1). The spores were digested for 25 min at 37°C, and a light microscope was used to confirm that the ascii were open. We quenched the digestion with 900 µl of sterile water, and washed them two times with 1 ml of sterile water. 1 ml of 2× YPD was added to the spores and they were grown at 30°C for 7 hr. 250,000 Mata cells were sorted into 5 ml polystyrene round-bottom test tubes (Falcon #352058). 1 ml 2× YNB with 2% glucose was added to the sorted cells in sheath fluid (PBS, Bio-Rad #12012932) and they were transferred to a shaking incubator and grown at 30°C. We sought to minimize the outgrowth of our cultures, and attempted to capture the culture at mid log in 2× YNB with sheath fluid the next morning. However, if the culture grew too much, we diluted the yeast in 1× YNB with 2% glucose to an OD600 of ~0.1 in a 50-ml flask. The culture was grown until the density reached an OD600 of ~0.5 and the yeast were harvested with vacuum filtration. For our experiments with the BY and RM cross (YLK3051), the cells were harvested in 2× YNB medium containing sheath fluid. For our other de novo eQTL experiments, the cells were captured as early as practicable. FACS was performed using a Bio-Rad S3e cell sorter with a cooled sample block.

For our allele-specific expression experiments, parental diploids were grown overnight in YPD medium and diluted in the morning to an OD600 of ~0.05 in 1× YNB. The diploid yeast were pooled in some experiments and for every experiment they were grown to a OD600 of between 0.4 and 0.6. The yeast were then harvested using vacuum filtration, flash frozen, and transferred to the –80°C freezer.

## Single-cell library preparation

Yeast cells stored on filters were removed from the –80°C freezer and the cells were fixed in 5 ml of 80% methanol. The yeast were placed in the –20°C freezer for 10 min. The fixed yeast were washed three times with 1 M Sorbitol. We partially digested the cell walls of the fixed yeast cells with Zymolyase 20T (Amsbio, #120491-1) to enable lysis in the Chromium device from 10× Genomics. In more detail, 200 µl of cells were combined with 0.5 µl of 500 U/ml of Zymolyase 20T and 1 µl of 10% Beta-Mercaptoethanol and digested for 20 min at 30°C with gentle shaking (250 rpm). For our diploid experiments, the zymolyase concentration was reduced to 250 U/ml. Phase microscopy was used to confirm that the digestion made partial spheroplasts. We washed digested yeast cells three times in 1 M Sorbitol with the centrifuge cooled to 4°C and after each wash step the yeast were spun at 500 × *g* for 5 min. The yeast were diluted to 1000 cells/µl and loaded onto the Chromium device using either the Single Cell 3′ Solution V2 or V3 kit. Library prep was carried out according to the manufacturer's protocol. Prepared libraries were sequenced on the Illumina Novaseq 6000 or Nextseq 2500. Each sequencing library was treated as a different 'batch' for all downstream analyses.

## scRNA-seq data processing

Sequencing reads were analyzed using *Cellranger* (version 5.01) using the S2888C reference genome (SGD, R64-2-1) (*Engel et al., 2014*). The transcriptome was amended to include the 3′ untranslated region (UTR) in the gene model using a custom python script (https://gist.github.com/theboocock/aacf72277a572ee3fe589c430bfd496e; *Boocock, 2023*). We obtained 3′ UTR lengths from an experimental dataset (*Xu et al., 2009*) and used the median 3′ UTR length for genes when this information was not available.

## Cell-cycle stage classification

We used unsupervised clustering based on cell-cycle gene expression, along with well-described marker gene expression, to classify yeast single cells into five different stages of the cell cycle (M/G1, G1, G1/S, S, and G2/M). Filtered gene expression matrices were loaded into the Seurat package (v4.04) (*Hao et al., 2021*) of the R programming language (v4.4.0) (*R Development Core Team, 2022*). 799 cell-cycle genes were obtained from previous cell-cycle synchronization experiments where microarrays were used to measure RNA levels (*Spellman et al., 1998*). Of these 799 genes, 787 were reliably quantified in our single-cell datasets and were used in the subsequent analysis. For each 10× library, we extracted the 787 cell-cycle genes from our filtered gene expression matrices and performed normalization using SCtransform (*Hafemeister and Satija, 2019*). We constructed a shared nearest neighbor graph using 12 principal components and identified clusters using the louvain algorithm set to a resolution of 0.3. We performed UMAP with 12 principal components to visualize these clusters in two dimensions. The Wilcoxon rank sum test, as implemented in the FindAllMarkers function of Seurat, was used to identify markers between the clusters. The identified markers were filtered to remove those with a $\log_2$ fold change of less than 0.2 and an adjusted p-value of less than 0.05. These markers were annotated with their cell-cycle classification from *Spellman et al., 1998*.

To classify haploid yeast into their cell-cycle stage, we used a list of 22 cell-cycle genes that are highly expressed in a certain stage and well-represented across our datasets (*Spellman et al., 1998*). For the M/G1 stage, we used the genes *PIR1*, *EGT2*, *ASH1*, *DSE1*, *DSE2*, and *CTS1*. For the G1 stage, we used the gene *MFA1*, which in our experiments reproducibly connected the M/G1 and G1/S transition stages. For the G1/S stage, we used the genes *CSI1*, *TOS4*, *POL30*, *PRY2*, *AXL2*, and *CLN2*. For the S stage, we used these genes *HTB1* and *HHF2*. Finally for the G2/M stage we used the genes *HOF1*, *PHO3*, *MMR1*, *CLB2*, *WSC4*, *CDC5*, and *CHS2*. We intersected these known markers with the differentially expressed transcripts identified with Seurat, as described above. Using this information, we manually classified every unsupervised cluster into one of the five stages of the cell cycle (M/G1, G1, G1/S, S, and G2/M). We could not identify a discrete G1 cell-cycle stage between M/G1 and G1/S

in our diploid single cell. We therefore classified our diploid single-cell data into four cell-cycle stages (M/G1, G1/S, S, and G2/M) using the same markers as above excluding *MFA1* which is not expressed in diploids.

## Single-cell variant counting and genotype inference

We obtained deep (>100×) paired-end sequenced data for our parental strains from previous work (*Bloom et al., 2019b*) and generated a variant call file (VCF) using the standard Genome Analysis Toolkit (GATK, v3.8.0) variant calling pipeline (*DePristo et al., 2011*). For each cross, we extracted biallelic SNPs segregating in each cross and used Vartrix version 1.0 (https://github.com/10XGenomics/vartrix; *10XGenomics, 2018*) to generate UMI counts for each parental allele from the cell ranger binary sequence alignment/map (BAM) file (*Li et al., 2009*). Doublet cells were identified by observing an excessive fraction of variant sites where we observed both parental alleles for a given cell barcode, and removed from downstream analyses. Counts at variants with extremely distorted allele frequencies (minor allele frequency <5%) were treated as missing and genotypes at those sites were imputed using the HMM described next.

We used an HMM to infer the genotypes of the recombinant progeny (*Broman, 2005*; *Arends et al., 2010*). The HMM is used to calculate the probability of underlying genotypes for each individual and requires three components: (1) prior probabilities for each of the possible genotypes, (2) emission probabilities for observing variant informative reads given each of the possible genotypes, and (3) transition probabilities – the probabilities of recombination occurring between adjacent genotype informative sites.

We defined prior genotype probabilities as 0.5 for each parental variant. Emission probabilities were calculated as previously described for low coverage sequencing data (*Dodds et al., 2015*; *Bilton et al., 2018*) under the assumption that the observed counts of reads for both possible variants (*Y*) at a genotype informative site (*g*) arise from a random binomial sampling of the alleles present at that site and that sequencing errors (*e*) occur independently between reads at a rate of 0.005:

$$p(Y \mid g = A) = \binom{D}{r} (1 - e)^r (1 - (1 - e))^{D-r}$$

$$p(Y \mid g = B) = \binom{D}{r} (e)^r (1 - e)^{D-r}$$

where *D* is the total read depth at a genotype informative site for a given individual, *r* is the total read depth for the A variant at that site, and A represents the variant from the haploid A parent and B represents the variant the haploid B parent. Transition probabilities were derived from existing genetic maps for the crosses (*Bloom et al., 2019a*). We linearly interpolated genetic map distances from the existing map to all genotype informative sites in our cross progeny.

For our combined single-cell datasets from each of our three crosses, we calculated the fraction of cells with unique genotypes by binarizing the genotype based on whether the genotype probability was greater than 50%. We then calculated the pairwise hamming distance between every pair of cells. Crosses with greater than 10% non-unique genotypes were further processed to retain the unique segregants with the highest UMI count. This filtering step was only needed for the cross of YJM981 and CBS2888, and the procedure reduced the number of cells from 14,823 to 6595.

## Local eQTL mapping in previously genotyped segregants

After inferring genotype probabilities, as described in the above section, the genotype probabilities for each single cell were correlated with the genotypes of existing segregants that were previously determined by whole-genome sequencing (*Bloom et al., 2013*). Segregant identity was determined by picking the previously genotyped segregant with genotypes that were most correlated with the genotype probability vector for a given single cell. Next we fit a negative binomial regression model that included a fixed effect of the natural log of total UMIs per cell (to control for compositional effects), a fixed effect for batch, a fixed effect for the genotypic marker closest to the transcript, and a random effect of segregant identity. Model parameters were estimated using iteratively reweighted least squares as implemented in the 'nebula' function in the Nebula R package (*He et al., 2021*) with

default arguments. p-values for the effect of the genotypic marker closest to the transcript were adjusted for multiple testing using the procedure of *Benjamini and Hochberg, 1995*.

## One-pot eQTL mapping

Genotype probabilities were standardized, and markers in very high LD ($r > 0.999$) were pruned. This LD pruning is approximately equivalent to using markers spaced 4 centimorgans (cm) apart. For each transcript, we counted the number of cells for which at least one UMI count was detected. Transcripts with non-zero counts in at least 128 cells were used for downstream analyses.

For each expressed transcript we first fit the negative binomial generalized linear model:

$$\mathbb{E}\left[Y\right] = \mu \tag{1}$$

$$Var(Y) = \mu + \frac{1}{\theta}\mu^2 \tag{2}$$

$$\mu = \exp(\beta_i + X_t\beta_t + \mathbf{X_b}\beta_b + X_c\beta_c) \tag{3}$$

which has the following log-likelihood:

$$\ell(\beta, \theta) = -\sum_{n=1}^{N}\left[(y_n + \theta)\log(\mu_n + \theta) - y_n\log(\mu_n) + \log(\left|\,\Gamma(y_n + 1)\,\right|) - \right.$$
$$\left.\theta\log(\theta) + \log(\left|\,\Gamma(\theta)\,\right|) - \log(\left|\,\Gamma(\theta + y_n)\,\right|)\right] \tag{4}$$

And where $Y$ is a vector of UMI counts per cell, $X_t$ is a vector of the natural logarithm of the total UMIs per cell and controls for compositional effects, $\mathbf{X_b}$ is an indicator matrix assigning cells to batches, and $X_c$ is the vector of standardized genotype probabilities across cells for the closest genotypic marker to each transcript from the pruned marker set. In addition, $\beta$ is a vector of estimated coefficients from the model, $\mu_n$ is the expected value of $Y$ for a given cell $n$, $N$ is the total number of cells, and $\theta$ is a negative binomial overdispersion parameter. Model parameters were estimated using iteratively reweighted least squares as implemented in the 'nebula' function in the Nebula R package (*He et al., 2021*). Due to the computational burden of fitting so many GLMs in the context of sc-eQTL mapping, we chose to estimate $\theta$ once for each transcript and use that estimate of $\theta$ in the additional models for that transcript within the cell-cycle stages, as described below. This approach is conservative, as the effects of unmodeled factors (for example *trans* eQTLs) will be absorbed into the estimate of overdispersion, resulting in larger estimated overdispersion ($\frac{1}{\theta}$) and lower model likelihoods. Computational approaches that re-estimate $\theta$ for each model, that jointly model all additive genetic effects, or that regularize $\theta$ across models and transcripts (*McCarthy et al., 2012*), may further increase statistical power to identify linkages.

To evaluate the statistical significance of local eQTLs, a likelihood ratio statistic $-2(\ell_{nc} - \ell_{fc})$, was calculated, comparing the log-likelihood of this model described above ($\ell_{fc}$) to the log-likelihood of the model where $\beta$ is re-estimated while leaving out the covariate $X_c$ for the closest marker to a transcript eQTL marker ($\ell_{nc}$). A permutation procedure was used to calculate FDR-adjusted p-values, and is described further below.

For each expressed transcript in each cell-cycle stage we also scanned the entire genome for eQTLs, enabling detection of *trans* eQTLs. A similar procedure was used as for the local eQTL-only scan except that *Equation 3* was replaced with:

$$\mu = \exp(\beta_i + X_t\beta_t + \mathbf{X_b}\beta_b + X_g\beta_g) \tag{5}$$

where $X_g$ is a vector of the scaled genotype probabilities at the gth genotypic marker, and the model is fit separately, one at a time, for each marker across the genome for each transcript. A likelihood ratio statistic for each transcript, within each cell-cycle stage, for each genotypic marker is calculated by comparing this model to the model where $\beta$ is re-estimated while leaving out the covariate. The likelihood ratio statistic was transformed into an LOD score, by dividing it by $2\log_e(10)$. We also used functions in the fastglm R package (*Huling, 2022*) for this scan, again re-using estimates of $\theta$ obtained as described above for each transcript across each cell-cycle stage. For each transcript and each $X_g$ chromosome, QTL peak markers were identified as the marker with the highest LOD score. The 1.5

LOD-drop procedure was used to define approximate 95% CIs for QTL peaks (*Dupuis and Siegmund, 1999*).

FDR-adjusted p-values were calculated for QTL peaks. They were calculated as the ratio of the number of transcripts expected by chance to show a maximum LOD score greater than a particular LOD threshold versus the number of transcripts observed in the real data with a maximum LOD score greater than that threshold, for a series of LOD thresholds ranging from 0.1 to 0.1+ the maximum observed LOD for all transcripts within a cell-cycle stage assignment, with equal-sized steps of 0.01. The number of transcripts expected by chance at a given threshold was calculated by permuting the assignments of segregant identity within each batch relative to segregant genotypes, calculating LOD scores for all transcripts across the chromosome as described above, and recording the maximum LOD score for each transcript. In each permutation instance, the permutation ordering was the same across all transcripts. We repeated this permutation procedure five times. Then, for each of the LOD thresholds, we calculated the average number of transcripts with maximum LOD greater than the given threshold across the five permutations. We used the 'approxfun' function in R to interpolate the mapping between LOD thresholds and FDR and estimate an FDR-adjusted p-value for each QTL peak (*Albert et al., 2018*). To detect QTL affecting transcript levels across cell-cycle stages and increase power to detect such effects, the same procedure was performed after summing the LODs across cell-cycle stage assignments and summing the LODs from permutations within cell-cycle stage assignments.

## Cell-cycle eQTL interactions

We tested for cell cycle by genotype interactions by calculating

$$Z = \frac{\beta_i - \beta_j}{\sqrt{(SE_i^2 + SE_j^2)}} \tag{6}$$

where $i$ and $j$ indicate two cell-cycle stages being contrasted. $\beta$ and $SE$ correspond to the QTL effect size and standard error from the modeling described above. The $Z$-statistic was assumed to come from a standard normal distribution, converted to a p-value, and FDR adjusted using the procedure of *Benjamini and Hochberg, 1995*. The adjustment was performed jointly either across all local eQTL for all transcripts for the local eQTL by cell-cycle interaction test, or jointly across all eQTL hotspots for all transcripts linking to that hotspot for the distant eQTL by cell-cycle interaction test.

## Allele-specific expression and overdispersion

Where multiple F1 hybrid diploids were assayed in the same experiment, we used a custom likelihood-based procedure to classify cells to one of the expected input diploids. For each F1 hybrid diploid, for transcripts where an allelic count was observed in at least 64 cells and for cells with less than 20,000 UMIs we tested for allele-specific effects on gene expression noise for each transcript separately using a negative binomial parameterization with the 'glmmTMB' function from the glmmTMB R package (*Brooks et al., 2017*).

Following the model description defined in the above section, 'One-pot eQTL mapping', with all terms not described here being the same as they are above, for each transcript *Equation 2* was replaced with:

$$Var(Y) = \mu + \frac{1}{\theta_k \mathbf{X_k}} \mu^2 \tag{7}$$

And *Equation 3* was replaced with:

$$\mu = \exp(\beta_i + \mathbf{Z}X_t\beta_t + \mathbf{ZX_b}\beta_b + \mathbf{X_k}\beta_k + \mathbf{ZX_l}\beta_l + \mathbf{ZX_l}X_k\beta_m) \tag{8}$$

where here $Y$ is a vector of all allelic counts from both alleles for a given transcript, stacked, $\mathbf{Z}$ is an indicator matrix mapping allelic counts to cells, $\mathbf{X}_k$ is an indicator matrix assigning counts to parental alleles, is a vector of the allele-specific dispersion effects, and $\mathbf{X}_l$ is an indicator matrix assigning cells to cell-cycle stages. The $t$-statistic for the allele-specific dispersion effect in the joint model was used to identify significant noise effects. Multiple testing corrected p-values were calculated for this statistic as described in the 'One-pot eQTL mapping section'.

## Noise terminology and overdispersion

Throughout the main text we refer to the overdispersion parameter estimate $\frac{1}{\theta}$ as 'noise'.

## Investigating bias in allele-specific expression and noise estimation

We investigated bias in the estimation of the overdispersion parameter by simulating counts from the above model, excluding all covariates except an intercept and allelic fold change from *Equation 8*. We kept the total number of cells fixed at 5000, and total expression per transcript was fixed at values equivalent to the median or the top 5% of expressed transcripts, reflecting observed values from the BY × RM F1 hybrid diploid. We simulated 250 instantiations each from a grid of allelic fold changes and dispersion fold changes that spanned our observed data, and refit the model. Results are shown in Figure S10. We note that p-values for the test for allelic effects on dispersion appear properly calibrated and non-significant despite the downward bias in overdispersion fold change estimates when alleles have very low counts.

## Accounting for the globally observed negative correlation between allele-specific noise and allele-specific average expression

We investigated the relationship between all allele-specific average expression effects and allele-specific noise effects across our F1 hybrids. We observed a negative correlation between allele-specific noise and allele-specific average expression. To identify allele-specific noise effects not explained by this relationship, we fit a robust linear regression model to the observed trend with the '*lmRob*' function from the robustbase package (v0.99-2) (*Maechler, 2023*) in data that was filtered to only include genes that had a significant allele-specific average expression effect and/or a significant allele-specific noise effect. We considered an allele-specific noise effect to violate the trend if the 95% CI of this noise effect did not overlap the 95% CI of the trend line.

## Cell-cycle occupancy mapping

We treated the assignment of cells to a given cell-cycle stage as separate binary traits. We mapped each binary trait using a logistic regression using the 'fastglmPure' function from the fastglm package (*Huling, 2022*). The effect of $\ln(UMIs)$ per cell was added as an additional covariate to control for unwanted technical effects of variable UMIs per cell. A family-wise error rate significance threshold was computed using the procedure of *Li and Ji, 2005*. Cell-cycle occupancy mapping was not performed on chromosome III due to linkage between the markers on that chromosome and the mating locus, which is highly distorted in allele frequency toward the MATa parent in our experiments.

## Hotspot identification, functional annotation, and localization

We sought to identify *trans*-acting eQTL hotspots in our one-pot eQTL experiments. To achieve this goal, we broke up the yeast genome into bins of 50 kilobases using the GenomicRanges package (v1.50.2) (*Lawrence et al., 2013*). We counted the number of distant eQTLs, defined here as transcripts physically not located on the same chromosome as the bin that had eQTL peaks overlapping the bin. We then asked whether under a Poisson model the number of linking transcripts could be expected by chance. We adjusted the p-value using a Bonferonni correction for the total number of bins. The mean of the Poisson distribution was taken to be the average number of transcripts linking to each bin. We merged significant bins if they were adjacent to each other giving us a final set of hotspots bins. We repeated this analysis on the previously obtained bulk RNA sequencing based eQTL mapping results (*Albert et al., 2018*) to enable us to compare our one-pot eQTL results from this cross to the results from bulk eQTL mapping, using consistent methodology.

We set out to annotate hotspots with functional information to identify candidate causal genes and variants. To approximate the CI of a hotspot, we extracted the 20 most significant eQTLs within each hotspot and took the 10th and 90th percentile left and right CI of these eQTL to be the hotspot CI. We conservatively extended the CI of each hotspot by two markers on either side. We intersected the CI of each hotspot with four different annotations: (1) causal genes from a previous QTL mapping study using segregants from the same crosses (*Bloom et al., 2019a*), (2) QTL CIs from the same study, (3) cell-cycle occupancy QTL from our study, and (4) genetic variants segregating in each cross. The functional consequences of these variants were generated with snpEFF (v5.1) (*Cingolani et al., 2012*). We added the allele frequency and PROVEAN (*Choi and Chan, 2015*) score of variants found in the

1,011 yeast genomes project (*Peter et al., 2018*). We performed GO (*Ashburner et al., 2000*) and Kyoto Encyclopedia of Genes and Genomics (*Kanehisa and Goto, 2000*) enrichment analyses of significant eQTLs within each hotspot using TopGO (v2.56.0) (*Alexa and Rahnenfuhrer, 2022*). These enrichments were calculated for the molecular function and biological process categories of the GO. The background for these enrichments were genes that were tested for eQTLs in each dataset and only genes that were not found on the same chromosome as the hotspot QTL. Fisher's exact test was used to calculate a p-value, any enrichment with a $\log_2$ fold change of greater than 2 and a nominal $p < 10^{-5}$ were output into a table for further inspection. These analyses were combined into a single worksheet, one for each hotspot, and we explored these worksheets to identify candidate causal genes and variants.

## *GPA1* allele-replacement strains

To generate allele replacement strains for the W82R and S469I variants of *GPA1*, we used a single-guide RNA CRISPR system to introduce double-strand breaks near our region of interest and provided coupled repair templates to replace the desired allele (*Sadhu et al., 2018*). The specifics of this procedure are described below.

We engineered a yeast strain derived from BY4741 (YLK3221; BY4741; Mata met15Δ his3Δ1 leu2Δ0 ura3Δ0 nej1Δ::KanMX) with all three natural combinations (82W 469S, 82W 469I, 82R 469S) of the W82R and S469I alleles. This strain contained a galactose inducible Cas9 (PLK77, p415-GalL-Cas9-CYC1t) (*DiCarlo et al., 2013*). Since BY4741 had the 469I allele, we first transformed a plasmid (PLK126) derived from PLK88 (SNR52p-gRNA(BstEII/SphI).CAN1.Y-SUP4t) (*Schubert et al., 2022*) designed to change the codon at that position to serine (S) – the variant commonly found in the population (chrVIII:113512_T/C). This plasmid was also designed to introduce a second edit in the 3′ UTR of *GPA1* (chrVIII:113496_T/C), which was needed to break the PAM site of the guide RNA used for editing. We grew the cells in galactose medium to induce the expression of Cas9, and confirmed that the mutation was incorporated into the genome of individual yeast using colony PCR and sanger sequencing (YLK3302 and YLK3303). To create the 82R mutation in these strains, we cured the yeast of the guide RNA plasmid and transformed another plasmid designed to change the codon at position 82 from tryptophan (W) to arginine (R) (PLK125). This plasmid introduces two edits, one at position (chrVIII:114674_A/G), this is the common variant that is found in the population, and another edit (chrVIII:114672_C/T) that makes a synonymous codon change and was needed to break the PAM site used for editing. We confirmed that these edits were incorporated into the genome of each yeast with Sanger sequencing (YLK3304 and YLK3305). We cured our allele-replacement strains of their guide RNA and Cas9 plasmids and obtained two colonies of each strain for use in subsequent phenotyping (YLK3306–YLK3311). We cured our parent strain (YLK3221) of the Cas9 plasmid and obtained two colonies for use in subsequent phenotyping (YLK3312–YLK3313).

## Single-cell sequencing of *GPA1* allele-replacement strains

Four biological replicates of allele replacement strains with the two natural allelic combinations of *GPA1* were individually pooled (YLK3306 and YLK3307 82W 469S, 420/421 82R 469S) and grown overnight in YPD at 30°C. These strains were then diluted to ~0.1 OD600 in a 250-ml flask containing 50 ml of YNB medium with a complete supplement mixture and allowed to grow for two doublings to an OD600 of ~0.5. The yeast were harvested using vacuum filtration. Samples were flash frozen in dry ice and ethanol and transferred to the –80°C freezer. Prior to loading the Chromium device, frozen cells were fixed in 80% methanol for 10 min, washed three times with sorbitol, and diluted to 1000 cells/µl. The 10× reagents were modified by removing 1 µl of beads and replacing it with 1 µl of zymolyase according to *Vermeersch et al., 2022*. Using these modified reagents, the Chromium device was loaded conventionally and all other aspects of library preparation and loading was done according to 10× and Illumina standard protocols.

## Growth measurements

All allele replacement strain growth experiments were performed at 30° in YP medium (2% bacto-peptone, 1% yeast extract) supplemented with 2% glucose using the approach described in *Boocock et al., 2021*. Strains were incubated with fast shaking in a Biotek synergy 2 plate reader. Before each experiment, strains were grown to saturation in our plate reader in 96-well plates

(Corning, Flat Bottom with Lid, #3370) in 2% glucose. Strains were then diluted 1:100 or transferred with a plastic 96-well pinner into new 96-well plates and transferred to a Bio-Tek Synergy plate reader, which automatically took optical density measurements (OD600) measurements every 15 min.

## Growth rate calculations

Growth rate was quantified as the geometric mean rate of growth (GMR). Our procedure for calculating the GMR follows that described in Brem et al. (*Roop et al., 2016*). Briefly, we fit a spline in R using the 'splinefun' function, and the time spent (*t*) between OD 0.2 and 0.8 was calculated. The GMR was then estimated as the log(0.8/0.2)/*t*. Plates were manually inspected for outliers and for those plates we retained any GMR greater than 0.07 and less than 0.085. This filter removed 20 data points out of 360 from further analysis. We converted the GMR of each well into doublings per hour. To determine whether our allele replacement strains changed the growth rate, we fit a linear model with an additive effect for genotype and plate. We used the emmeans package (v1.10.1) (*Lenth, 2023*) to extract pairwise contrasts and performed multiple-hypothesis correction with the Tukey method. For visualization purposes, we extracted the residuals from a model with the additive plate effect and added the intercept from this model to these residuals. We created a boxplot of these residuals split by genotype with the ggplot2 (v3.5.1) package (*Wickham, 2009*).

## Mating efficiency experiments

We performed competitive mating assays to test whether our *GPA1* allele replacement strains (YLK3306 82W 469S, YLK3308 82R 469S, YLK3312 82W 469I) altered the efficiency of mating. Because our allele replacement strains were isogenic except for the engineered variants, we adapted a classic yeast mating assay to use fluorescence instead of selectable markers (*Sprague, 1991*). We transformed each of our MATa allele replacement strains with a constitutively expressed mTurquoise (blue) and mRuby2 (red) fluorescent marker on a HIS3 expressing 2 µm plasmid created with the Moclo yeast cloning toolkit (YLK3314–YLK3319) (*Lee et al., 2015*). Each strain was transformed independently with both plasmids; this was done to ensure that we could control for any possible effect of the fluorescent protein on mating efficiency.

We first grew up pairs of strains with combinations of fluorescent markers for 2 days in selective medium (YNB complete -his + 2% glucose). We grew up the mating tester strain (YLK3218, BY4742; MATα leu2Δ his3Δ ura3Δ lys2Δ) overnight in complete minimal medium (YPD +2% glucose). We mixed 2 ml containing $10^6$ cells from both of the allele replacement strains. We plated 200 µl of the mix on two agar plates containing selective medium at a high density. These plates were used to estimate the ratio of strains in the mix before performing the mating assay. We combined 2 ml of the mix with 1 mLcontaining $10^7$ cells from the mating tester strain. The 3 ml mix of all three strains was filtered through a 125-ml vacuum filtration system (Sigma-Aldrich #Z290467) fitted with a 0.2 µM nylon membrane filters (Sterlitech, #NY0225100). We placed the filter on a YPD agar plate facing upwards to facilitate mating between the MATa and MATα yeast strains. After 4 hr, we placed the cells in a 50-ml conical tube containing 1 ml of water and washed the yeast off the filter. We transferred the yeast onto 3 agar plates that select for the plasmid and diploids (YNB minimal + leu + ura + 2% glucose). These plates were used to estimate the ratio of strains after mating had occurred. The pre- and post-mating plates were then allowed to incubate for 3 days at 30°C.

We used flow cytometry to measure the ratio of fluorescence before and after mating with a Bio-Rad S3e cell sorter. The FSC, SSC, and fluorescence gates were calibrated using one of the pre-mating samples, and they remained fixed for other samples. $10^6$ events were captured from each sample. The pre-mating values were averaged and subtracted from the post-mating values. To calculate the mating efficiency, we assumed that the wild-type strain had an efficiency of 100% and we normalized every sample to the average of the wild-type strain for each color. We used linear regression to determine whether the allele replacement strains altered mating efficiency. We used the emmeans package (v1.10.1) (*Lenth, 2023*) to extract pairwise contrasts and performed multiple-hypothesis correction with the Tukey method. This whole experimental procedure was repeated once more on a different day; and, in total, each strain was grown up four times, twice for each genotype and fluorescent plasmid combination.

## Population genetics analysis

We downloaded the reads for the 1011 yeast strains from Peter et al. from the short-read archive and generated a VCF using the standard Genome Analysis Toolkit (GATK, v4.2.0) variant calling pipeline (*DePristo et al., 2011*). Variants were filtered using bcftools (v1.15) (*Danecek et al., 2021*) to include any sites with a mapping quality (MQ) greater than 40, mapping quality rank sum (MQRankSum) greater than –12.5, read position rank sum (ReadPosRankSum) greater than –8, quality by depth (QD) greater than 2, total depth less than 986,842.5, variant quality greater than 100, and fraction missing (F_MISS) less than 10%. We further filtered our VCF to only include biallelic SNP sites with a population frequency of greater than 5%. We made a dissimilarity matrix using SNPRelate (v1.38.0) (*Zheng et al., 2012*) and built a neighbor-joining tree with ape (v5.8) (*Paradis and Schliep, 2019*) and visualized the tree with the ggtree package (v3.12.0) (*Yu et al., 2017*).

To assess whether the W82R variant was enriched in mosaic clades of yeast as defined by *Peter et al., 2018*, we extracted all variants in the population with an allele frequency of between 18.5% and 22.5%, 2% plus or minus the W82R variant allele frequency of 20.5%. There are 8136 variants with this frequency in the population. We tested whether the allele frequency of the W82R in the mosaic clades was significantly enriched in mosaic clades of yeast by randomly sampling with replacement these variants 10,000 times and estimating their allele frequency in the mosaic clades. The number of times the allele frequency of these variants was greater than the W82R variant was used to generate a permutation p-value.

## Code availability

Code for the HMM, eQTL mapping, and gene expression noise analysis can be found at https://github.com/joshsbloom/single_cell_eQTL/tree/master/yeast/code (copy archived at *Bloom, 2025*). Data and code to recreate the figures in the manuscript can be found at https://github.com/theboocock/yeast_single_cell_post_mapping_analysis (copy archived at *Boocock, 2024b*). This repository also contains the code that performs *trans*-eQTL hotspot analysis, cell-cycle stage assignment, raw data processing, and additional links to generated data.

## Acknowledgements

We thank Frank Albert, Stefan Zdraljevic, and Giancarlo Bruni for helpful manuscript feedback and edits. We thank Eyal Ben-David and Longhua Guo for helpful discussions during the early stages of the project. This work was supported by funding from the Howard Hughes Medical Institute (to LK) and NIH grant 2RO1GM102308-06 (to LK).

## Additional information

### Funding

| Funder | Grant reference number | Author |
| --- | --- | --- |
| National Institutes of Health | 2RO1GM102308-06 | Leonid Kruglyak |
| Howard Hughes Medical Institute | | Leonid Kruglyak |

The funders had no role in study design, data collection and interpretation, or the decision to submit the work for publication.

### Author contributions

James Boocock, Conceptualization, Resources, Data curation, Software, Formal analysis, Validation, Investigation, Visualization, Methodology, Writing – original draft, Writing – review and editing; Noah Alexander, Conceptualization, Software, Investigation, Methodology; Leslie Alamo Tapia, Validation, Investigation, Methodology; Laura Walter-McNeill, Chetan Munugala, Validation; Shivani Prashant Patel, Validation, Investigation; Joshua S Bloom, Leonid Kruglyak, Conceptualization, Resources, Data curation, Software, Formal analysis, Supervision, Funding acquisition, Validation, Investigation,

Visualization, Methodology, Writing – original draft, Project administration, Writing – review and editing

## Author ORCIDs
James Boocock (ID) https://orcid.org/0000-0003-0323-8818
Joshua S Bloom (ID) https://orcid.org/0000-0002-7241-1648
Leonid Kruglyak (ID) https://orcid.org/0000-0002-8065-3057

Reviewer #1 (Public review): https://doi.org/10.7554/eLife.95566.3.sa1
Reviewer #2 (Public review): https://doi.org/10.7554/eLife.95566.3.sa2
Author response https://doi.org/10.7554/eLife.95566.3.sa3

# Additional files

## Supplementary files

Supplementary file 1. Yeast strains (Table S1), plasmids (Table S2), and primers used in this study (Table S3); summary information for the single-cell expression data generated in this study (Table S4); variance explained by cell-cycle stage and the effect of segregant per transcript for the 393 previously generated segregants (Table S5); and Local eQTL summary statistics for the 393 previously generated segregants (Table S6).

Supplementary file 2. Local eQTL summary statistics for our one-pot eQTL experiments. Each sheet has the local eQTL summary statistics for the three crosses we examined (Table S1 cross A=BY and RM, Table S2 cross B=YJM145 and YPS163, Table S3 cross C=CBS2888xYJM981). The column 'has cell-cycle interaction' is set to 1 if a cell-cycle interaction was observed at a FDR of <5%. For cross A, the summary statistics from bulk eQTL are provided in additional columns. For cross A, the summary statistics from bulk eQTL[7] are provided in additional columns. Missing values in the bulk eQTL columns indicate that a gene did not pass our filtering criteria, and a local eQTL test was not performed.

Supplementary file 3. Distant eQTL summary statistics for our one-pot eQTL experiments. Each sheet has the local eQTL summary statistics for the three crosses we examined (Table S1 cross A=BY and RM, Table S2 cross B=YJM145 and YPS163, Table S3 cross C=CBS2888xYJM981). The column 'has cell-cycle interaction' is set to 1 if a cell-cycle interaction was observed at a FDR of <5%. The column 'eQTL in hotspot' is set to 1 if a distal eQTL falls within a significant hotspot bin.

Supplementary file 4. Hotspot annotation files for each cross (cross A = BY and RM, cross B = YJM145 and YPS163, cross C = CBS2888xYJM981, A_bulk = BY and RM with expression measured by bulk RNA-seq *Albert et al., 2018*).

Supplementary file 5. Single-cell allele-specific expression (ASE) summary statistics. Each sheet has the ASE summary statistics for the parental hybrids of the three crosses we examined (Table S1 cross A=BY and RM, Table S2 cross B=YJM145 and Table S3 YPS163, cross C=CBS2888xYJM981). The local eQTL summary statistics were included from the corresponding one-pot eQTL experiment for each transcript that was tested there. Missing values in the one-pot eQTL columns indicate that a gene did not pass our filtering criteria, and a local eQTL test was not performed.

Supplementary file 6. Summary statistics for allele-specific effects on noise and average expression. Each sheet has the summary statistics for the parental hybrids of the three crosses we examined (Table S1 cross A=BY and RM, Table S2 cross B=YJM145 and YPS163, Table S4 cross C=CBS2888xYJM981). The estimates and p-values are derived from the joint model that contains cell-cycle, cell-cycle interactions, and allelic effects on both the mean and noise. The column 'Overlaps global trend line' is set to 1 if the 95% confidence interval of the noise effect did not overlap the 95% confidence interval of the global trend line.

Supplementary file 7. Cell-cycle occupancy QTL summary statistics for our one-pot eQTL experiments. Each sheet has the cell-cycle occupancy QTL summary statistics for the three crosses we examined (Table S1 cross A=BY and RM, Table S2 cross B=YJM145 and YPS163, Table S3 cross C=CBS2888xYJM981).

Supplementary file 8. Alignment and tree files from a clustered blast search of the yeast Gpa1 protein sequence (YHR005C).

Supplementary file 9. Single-cell RNA sequencing of allele-replacement strains with the 82R and 82W alleles of GPA1. The distant eQTL effects near GPA1 from our one-pot eQTL experiment are

contrasted to a differential expression analysis of our allele replacement strains. The p-values from our one-pot eQTL study were adjusted using a permutation procedure, and a Bonferroni correction was used to adjust the single-cell validation p-values. Missing values in the single-cell validation columns indicate that a gene did not pass our filtering criteria, and a differential expression test was not performed.

MDAR checklist

## Data availability

Sequencing data is available under the NCBI BioProject PRJNA1049497. Raw and processed data can be found at the locations https://doi.org/10.5061/dryad.xgxd254qb and https://zenodo.org/doi/10.5281/zenodo.12695127, respectively. Finemapping spreadsheets can be found on GitHub at https://github.com/theboocock/finemapping_spreadsheets_single_cell (copy archived at *Boocock, 2024a*).

The following datasets were generated:

| Author(s) | Year | Dataset title | Dataset URL | Database and Identifier |
|---|---|---|---|---|
| Bloom J | 2024 | Data for: Raw count data, transcribed variant count data, and reference genomic annotation files for Boocock et al. 2024 | https://doi.org/10.5061/dryad.xgxd254qb | Dryad Digital Repository, 10.5061/dryad.xgxd254qb |
| Boocock J | 2024 | Single-cell eQTL mapping in yeast reveals a tradeoff between growth and reproduction | https://doi.org/10.5281/zenodo.12695127 | Zenodo, 10.5281/zenodo.12695127 |
| Boocock J | 2024 | Single-cell eQTL mapping in yeast reveals a tradeoff between growth and reproduction | https://www.ncbi.nlm.nih.gov/bioproject/PRJNA1049497 | NCBI BioProject, PRJNA1049497 |

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

# Appendix 1

## Appendix 1—key resources table

| Reagent type (species) or resource | Designation | Source or reference | Identifiers | Additional information |
|---|---|---|---|---|
| Strain, strain background (*Saccharomyces cerevisiae*) | YLK3301 | *Bloom et al., 2019a* | | YPS163 MatA hoΔ::HphMX flo8Δ::NatMX (YLK2438) x YJM145 MatAlpha hoΔ::HphMX flo8Δ::NatMX (YLK2436) |
| Strain, strain background (*Saccharomyces cerevisiae*) | YLK3004 | *Bloom et al., 2019a* | | YJM981 MatAlpha hoΔ::HphMX (yST191) x CBS2888 MatA hoΔ::KanMX (Box A1 C3) |
| Strain, strain background (*Saccharomyces cerevisiae*) | YLK3051 | *Bloom et al., 2019a* | | BY MatA (YLK1879) x RM MatAlpha AMN1-BY hoΔ::HphMX flo8Δ::NatMX (YLK1950) |
| Strain, strain background (*Saccharomyces cerevisiae*) | YLK1993 | *Albert et al., 2018* | | BY MatA (YLK1879) x RM MatAlpha AMN1-BY hoΔ::HphMX flo8Δ::NatMX (YLK1950) |
| Strain, strain background (*Saccharomyces cerevisiae*) | YLK3221 | *Sadhu et al., 2018* | | Mata met15Δ his3Δ1 leu2Δ0 ura3Δ0 nej1Δ::KanMX Gpa1-82W,469S [p415 GalL-Cas9-Cyc1t] |
| Strain, strain background (*Saccharomyces cerevisiae*) | YLK3302 | This paper | | Mata chrVII:113512_C chrVIII:113496_C Gpa1-469S [p415 GalL-Cas9-Cyc1t] |
| Strain, strain background (*Saccharomyces cerevisiae*) | YLK3303 | This paper | | Mata chrVII:113512_C chrVIII:113496_C Gpa1-469S [p415 GalL-Cas9-Cyc1t] |
| Strain, strain background (*Saccharomyces cerevisiae*) | YLK3304 | This paper | | Mata chrVII:113512_C chrVIII:113496_C chrVIII:114674_G chrVIII:114672_T Gpa1-469S Gpa1-82W [p415 GalL-Cas9-Cyc1t] |
| Strain, strain background (*Saccharomyces cerevisiae*) | YLK3305 | This paper | | Mata chrVII:113512_C chrVIII:113496_C chrVIII:114674_G chrVIII:114672_T Gpa1-469S Gpa1-82W [p415 GalL-Cas9-Cyc1t] |
| Strain, strain background (*Saccharomyces cerevisiae*) | YLK3306 | This paper | | Mata chrVII:113512_C chrVIII:113496_C Gpa1-469S |
| Strain, strain background (*Saccharomyces cerevisiae*) | YLK3307 | This paper | | Mata chrVII:113512_C chrVIII:113496_C Gpa1-469S |
| Strain, strain background (*Saccharomyces cerevisiae*) | YLK3308 | This paper | | Mata chrVII:113512_C chrVIII:113496_C chrVIII:114674_G chrVIII:114672_T Gpa1-469S Gpa1-82W |
| Strain, strain background (*Saccharomyces cerevisiae*) | YLK3309 | This paper | | Mata chrVII:113512_C chrVIII:113496_C chrVIII:114674_G chrVIII:114672_T Gpa1-469S Gpa1-82W |
| Strain, strain background (*Saccharomyces cerevisiae*) | YLK3310 | This paper | | Mata chrVII:113512_C chrVIII:113496_C chrVIII:114674_G chrVIII:114672_T Gpa1-469S Gpa1-82W |
| Strain, strain background (*Saccharomyces cerevisiae*) | YLK3311 | This paper | | Mata chrVII:113512_C chrVIII:113496_C chrVIII:114674_G chrVIII:114672_T Gpa1-469S Gpa1-82W |
| Strain, strain background (*Saccharomyces cerevisiae*) | YLK3312 | This paper | | Mata met15Δ his3Δ1 leu2Δ0 ura3Δ0 nej1Δ::KanMX Gpa1-82W, Gpa1-469I |
| Strain, strain background (*Saccharomyces cerevisiae*) | YLK3313 | This paper | | Mata met15Δ his3Δ1 leu2Δ0 ura3Δ0 nej1Δ::KanMX Gpa1-82W Gpa1-469I |
| Strain, strain background (*Saccharomyces cerevisiae*) | YLK3314 | This paper | | Mata chrVII:113512_C chrVIII:113496_C Gpa1-469S [PLK127] |
| Strain, strain background (*Saccharomyces cerevisiae*) | YLK3315 | This paper | | Mata chrVII:113512_C chrVIII:113496_C Gpa1-469S [PLK128] |
| Strain, strain background (*Saccharomyces cerevisiae*) | YLK3316 | This paper | | Mata chrVII:113512_C chrVIII:113496_C chrVIII:114674_G chrVIII:114672_T Gpa1-469S Gpa1-82W [PLK127] |
| Strain, strain background (*Saccharomyces cerevisiae*) | YLK3317 | This paper | | Mata chrVII:113512_C chrVIII:113496_C chrVIII:114674_G chrVIII:114672_T Gpa1-469S Gpa1-82W [PLK128] |
| Strain, strain background (*Saccharomyces cerevisiae*) | YLK3318 | This paper | | Mata met15Δ his3Δ1 leu2Δ0 ura3Δ0 nej1Δ::KanMX Gpa1-82W, Gpa1-469S [PLK127] |
| Strain, strain background (*Saccharomyces cerevisiae*) | YLK3319 | This paper | | Mata met15Δ his3Δ1 leu2Δ0 ura3Δ0 nej1Δ::KanMX Gpa1-82W, Gpa1-469S [PLK128] |
| Recombinant DNA reagent | MF2 p41 neo (plasmid) | *Treusch et al., 2015* | RRID:Addgene_58564 | Flourescent magic marker plasmid with KanMX resistant cassette |

*Appendix 1 Continued on next page*

*Appendix 1 Continued*

| Reagent type (species) or resource | Designation | Source or reference | Identifiers | Additional information |
|---|---|---|---|---|
| Recombinant DNA reagent | MF2 p41 nat (plasmid) | *Treusch et al., 2015* | RRID:Addgene_58546 | Flourescent magic marker plasmid with NatMX resistant cassette |
| Recombinant DNA reagent | p415 GalL-Cas9-Cyc1t (plasmid) | *DiCarlo et al., 2013* | RRID:Addgene_43804 | Gal inducible CAS9 with LEU cassette |
| Recombinant DNA reagent | SNR52p-gRNA(BstEII/SphI).CAN1.Y-SUP4t (plasmid) | *DiCarlo et al., 2013* | RRID:Addgene_98814 | Guide RNA expression plasmid with URA resistance |
| Recombinant DNA reagent | plk88+GPA1 novel variant (plasmid) | This paper | | Guide RNA and coupled repair template to change 82 W to 82 R in Gpa1 |
| Recombinant DNA reagent | plk88+GPA1 reversion (plasmid) | This paper | | Guide RNA and coupled repair template to change 82I to 82 S in Gpa1 |
| Recombinant DNA reagent | HIS3 2 um with ruby2 (plasmid) | This paper | | ConLS-pTef1-mRuby2-tEno1-ConR1-His3-2micron-AmpR |
| Recombinant DNA reagent | HIS3 2 um with mTurquoise (plasmid) | This paper | | ConLS-pTef1-mTurquoise-tEno1-ConR1-His3-2micron-AmpR |
| Commercial assay or kit | Chromium Single Cell 3' v3 | 10 x Genomics | 10 X:CG000201 | |
| Software, algorithm | HMM, eQTL mapping, and noise analysis code | This paper | | Avaliable at https://github.com/joshsbloom/single_cell_eQTL, archived at: https://doi.org/10.5281/zenodo.14834926 |
| Software, algorithm | 3' UTR extension script for cell ranger | This paper | | Available at https://gist.github.com/theboocock/aacf72277a572ee3fe589c430bfd496e |
| Software, algorithm | Figure creation code | This paper | | Avaliable at https://github.com/theboocock/yeast_single_cell_post_mapping_analysis, archived at: https://doi.org/10.5281/zenodo.14834916 |

