## [Editor Report · eLife assessment]

This manuscript describes the mapping of natural DNA sequence variants that affect gene expression and its noise, as well as cell cycle timing, using as input single-cell RNA-sequencing of progeny from crosses between wild yeast strains. The method represents an **important** advance in the study of natural genetic variation. The findings, especially given the follow-up validation of the phenotypic impact of a mapped locus of major effect, provide **convincing** support for the rigor and utility of the method.

---

## [Referee Report · Reviewer #1 (Public review)]

The authors demonstrate that it is possible to carry out eQTL experiments for the model eukaryote *S. cerevisiae*, in "one pot" preparations, by using single-cell sequencing technologies to simultaneously genotype and measure expression. This is a very appealing approach for investigators studying genetic variation in single-celled and other microbial systems, and will likely inspire similar approaches in non-microbial systems where comparable cell mixtures of genetically heterogeneous individuals could be achieved.

While eQTL experiments have been done for nearly two decades (the corresponding author's lab are pioneers in this field), this single-cell approach creates the possibility for new insights about cell biology that would be extremely challenging to infer using bulk sequencing approaches. The major motivating application shown here is to discover cell occupancy QTL, i.e. loci where genetic variation contributes to differences in the relative occupancy of different cell cycle stages. The authors dissect and validate one such cell cycle occupancy QTL, involving the gene GPA1, a G-protein subunit that plays a role in regulating the mating response MAPK pathway. They show that variation at GPA1 is associated with proportional differences in the fraction of cells in the G1 stage of the cell cycle. Furthermore, they show that this bias is associated with differences in mating efficiency.

---

## [Referee Report · Reviewer #2 (Public review)]

Boocock and colleagues present an approach whereby eQTL analysis can be carried out by scRNA-Seq alone, in a one-pot-shot experiment, due to genotypes being able to be inferred from SNPs identified in RNA-Seq reads. This approach obviates the need to isolate individual spores, genotype them separately by low-coverage sequencing, and then perform RNA-Seq on each spore separately. This is a substantial advance and opens up the possibility to straightforwardly identify eQTLs over many conditions in a cost-efficient manner. Overall, I found the paper to be well-written and well-motivated, and have no issues with either the methodological/analytical approach (though eQTL analysis is not my expertise), or with the manuscript's conclusions.

---

## [Author Response]

The following is the authors’ response to the original reviews.

**Public Reviews:**

**Reviewer #1 (Public Review):**
Summary:The authors demonstrate that it is possible to carry out eQTL experiments for the model eukaryote *S. cerevisiae*, in "one pot" preparations, by using single-cell sequencing technologies to simultaneously genotype and measure expression. This is a very appealing approach for investigators studying genetic variation in single-celled and other microbial systems, and will likely inspire similar approaches in non-microbial systems where comparable cell mixtures of genetically heterogeneous individuals could be achieved.Strengths:While eQTL experiments have been done for nearly two decades (the corresponding author's lab are pioneers in this field), this single-cell approach creates the possibility for new insights about cell biology that would be extremely challenging to infer using bulk sequencing approaches. The major motivating application shown here is to discover cell occupancy QTL, i.e. loci where genetic variation contributes to differences in the relative occupancy of different cell cycle stages. The authors dissect and validate one such cell cycle occupancy QTL, involving the gene GPA1, a G-protein subunit that plays a role in regulating the mating response MAPK pathway. They show that variation at GPA1 is associated with proportional differences in the fraction of cells in the G1 stage of the cell cycle. Furthermore, they show that this bias is associated with differences in mating efficiency.Weaknesses:While the experimental validation of the role of GPA1 variation is well done, the novel cell cycle occupancy QTL aspect of the study is somewhat underexploited. The cell occupancy QTLs that are mentioned all involve loci that the authors have identified in prior studies that involved the same yeast crosses used here. It would be interesting to know what new insights, besides the "usual suspects", the analysis reveals. For example, in Cross B there is another large effect cell occupancy QTL on Chr XI that affects the G1/S stage. What candidate genes and alleles are at this locus? And since cell cycle stages are not biologically independent (a delay in G1, could have a knock-on effect on the frequency of cells with that genotype in G1/S), it would seem important to consider the set of QTLs in concert.

We thank the reviewer for this suggested clarification. We have modified the text to make it clear that cell cycle occupancy is a compositional phenotype. Like the reviewer, we also noticed the distal *trans* eQTL hotspot on Chr XI in Cross B, but we were not able to identify compelling candidate gene(s) or variant(s) despite extensive effort.

**Reviewer #2 (Public Review):**
Boocock and colleagues present an approach whereby eQTL analysis can be carried out by scRNA-Seq alone, in a one-pot-shot experiment, due to genotypes being able to be inferred from SNPs identified in RNA-Seq reads. This approach obviates the need to isolate individual spores, genotype them separately by low-coverage sequencing, and then perform RNA-Seq on each spore separately. This is a substantial advance and opens up the possibility to straightforwardly identify eQTLs over many conditions in a cost-efficient manner. Overall, I found the paper to be well-written and well-motivated, and have no issues with either the methodological/analytical approach (though eQTL analysis is not my expertise), or with the manuscript's conclusions.I do have several questions/comments.393 segregant experiment:For the experiment with the 393 previously genotyped segregants, did the authors examine whether averaging the expression by genotype for single cells gave expression profiles similar to the bulk RNA-Seq data generated from those genotypes? Also, is it possible (and maybe not, due to the asynchronous nature of the cell culture) to use the expression data to aid in genotyping for those cells whose genotypes are ambiguous? I presume it might be if one has a sufficient number of cells for each genotype, though, for the subsequent one-pot experiments, this is a moot point.

As mentioned in our preliminary response, while it is possible to expand the analysis along these lines, this is not relevant for the subsequent one-pot experiments. We have made all the data available so that anyone interested can try these analyses.

Figure 1B:Is UMAP necessary to observe an ellipse/circle - I wouldn't be surprised if a simple PCA would have sufficed, and given the current discussion about whether UMAP is ever appropriate for interpreting scRNA-Seq (or ancestry) data, it seems the PCA would be a preferable approach. I would expect that the periodic elements are contained in 2 of the first 3 principal components. Also, it would be nice if there were a supplementary figure similar to Figure 4 of Macosko et al (PMID 26000488) to indeed show the cell cycle dependent expression.

We have added two new figures (S2 and S3) that represent alternative visualizations of the cell-cycle that are not dependent on UMAP. Figure S2 shows plots of different pairs of principal components, with each cell colored by its assigned cell-cycle stage. We do not observe a periodic pattern in the first 3 principal components as the reviewer expected, but when we explore the first 6 principal components, we see combinations of components that clearly separate the cell cycle clusters. We emphasize that the clusters were generated using the Louvain algorithm and assigned to cell-cycle stages using marker genes, and that UMAP was used only for visualization.

We could not create a figure similar to Macosko et al. because of differences between the cell cycle categories we used and those of Spellman et al (PMID 9843569). We instead created Figure S3 to address the reviewer's comment. This figure uses a heatmap in a style similar to that of Macosko et al. to display cell-cycle-dependent expression of the 22 genes we used as cell cycle markers across each of the five cell cycle stages (M/G1, G1, G1/S, S, G2/M).

We have renumbered the supplementary figures after incorporating these two additional supplementary figures into the manuscript.

Aging, growth rate, and bet-hedging:The mention of bet-hedging reminded me of Levy et al (PMID 22589700), where they saw that Tsl1 expression changed as cells aged and that this impacted a cell's ability to survive heat stress. This bet-hedging strategy meant that the older, slower-growing cells were more likely to survive, so I wondered a couple of things. It is possible from single-cell data to identify either an aging, or a growth rate signature? A number of papers from David Botstein's group culminated in a paper that showed that they could use a gene expression signature to predict instantaneous growth rate (PMID 19119411) and I wondered if (a) this is possible from single-cell data, and (b) whether in the slower growing cells, they see markers of aging, whether these two signatures might impact the ability to detect eQTLs, and if they are detected, whether they could in some way be accounted for to improve detection.

As mentioned in our preliminary response, we are not sure how to look for gene expression signatures of aging in yeast scRNA-seq data. We believe that the proposed analyses are beyond the scope of the current paper. As noted above, we have made all the data available so that anyone interested can explore these hypotheses.

AIL vs. F2 segregants:I'm curious if the authors have given thought to the trade-offs of developing advanced intercross lines for scRNA-Seq eQTL analysis. My impression is that AIL provides better mapping resolution, but at the expense of having to generate the lines. It might be useful to see some discussion on that.

We thank the reviewer for the comments. We believe that a discussion of trade-offs between different approaches for constructing mapping populations, such as AIL and F2 segregants, is beyond the scope of this paper.

10x vs SPLit-Seq10x is a well established, but fairly expensive approach for scRNA-Seq - I wondered how the cost of the 10x approach compares to the previously used approach of genotyping segregants and performing bulk RNA-Seq, and how those costs would change if one used SPLiT-Seq (see PMID 38282330).

We thank the reviewer for the comments. We believe that a discussion of cost trade-offs between 10x and other approaches is beyond the scope of this paper, especially given the rapidly evolving costs of different technologies.

**Recommendations for the authors:**

**Reviewer #1 (Recommendations For The Authors):**
Throughout the results section the authors point to File S1 for additional information. This file is a tarball with about 20 Excel documents in it, each with several sheets embedded. The authors should provide a detailed README describing how to understand the organizations of the files in File S1 and the many embedded sheets in each file. Statements made in the manuscript about File S1 should explicitly direct the reader to a specific spreadsheet and table to refer to.

We have added an additional README file to the tarball that explains the organization of File S1 and describes the data contained in each sheet. Throughout the text, we now reference specific spreadsheets to assist the reader. In addition, these spreadsheets have been added to a github repository https://github.com/theboocock/finemapping_spreadsheets_single_cell

Neither of the two GitHub repositories referenced under "Code availability" has adequate documentation that would allow a reader to try and reproduce the analyses presented here. The one entitled https://github.com/joshsbloom/single_cell_eQTL has no functional README, while https://github.com/theboocock/yeast_single_cell_post_analysis is somewhat better but still hard to navigate. Basic information on expected inputs, file formats, file organization, output types, and formats, etc. is required to get any of these pipelines to run and should be provided at a minimum.

We thank the reviewer for the comment. In response, we have refactored both GitHub repositories and added extensive documentation to improve usability. We updated the versions of software and packages, this has been reflected in the methods section.

*S. cerevisiae* strains are preferentially diploid in nature and many genes involved in the mating pathway are differentially regulated in diploids vs haploids. Have the authors explored the fitness effects of the GPA1 82R allele in diploids? What is the dominance relationship between 82W and 82R?

We thank the reviewer for the comment. In diploid yeast, the mating pathway is repressed, and thus we would not expect there to be any fitness consequences due to the presence of different alleles of *GPA1*.

The diploid expression profiling (page 5 and Table S9) doesn't implicate GPA1; can you the authors comment on this in light of their finding in haploids?

The mating pathway, including *GPA1*, is repressed in diploids, and hence the expression of *GPA1* cannot be studied in these strains (PMID: 3113739). In addition, allele-specific expression differences only identify *cis*-regulatory effects. We know that the *GPA1* variant results in a protein-coding change, which may or may not influence the levels of mRNA in *cis*, so that even if GPA1 were expressed in diploids, there would be no expectation of an allele-specific difference in expression.

With respect to the candidate CYR1 QTL -- note that strains with compromised Cyr1 function also generally show increased sporulation rates and/or sporulation in rich media conditions (cAMP-PKA signaling represses sporulation). Is this the case in diploids with the CBS2888 allele at CYR1? If the CBS2888 allele is a CYR1 defect one might expect reduced cAMP levels. It is possible to estimate adenylate cyclase levels using a fairly straightforward ELISA assay. This would provide more convincing evidence of the causal mechanism of the alleles identified.

We thank the reviewer for the comment, and we agree that a functional study of the *CYR1* alleles would provide more convincing evidence for the causal mechanism of the connection between cell cycle occupancy, cAMP levels, and growth. However, we believe that the proposed experiments are beyond the scope of our current study. The evidence we provide is sufficient to establish that *CYR1* is a strong candidate gene for the eQTL hotspot.

Re: CYR1 candidate QTL -- The authors should reference the work of Patrick Van Dijck and Johan M Thevelein on CYR1 allelic variation, and other papers besides the Matsumoto/ Ishikawa papers, as the effects of cAMP-PKA signaling on stress can be quite variable. cAMP pathway variants, including in CYR1, have popped up in quite a few other yeast QTL mapping and experimental evolution papers. These should be referenced as well.

We thank the reviewer for these references; we have added a comment about the relationship between stress tolerance and *CYR1* variation, and cited the relevant references accordingly.

Figure S10 - the subfigure showing the frequency of the GPA 82R compared to 82W suggests a fairly large and deleterious fitness effect of this allele; on the order of 7-8% fewer cells per cell cycle stage than the 82W allele. Can the authors reconcile this with the more modest growth rate effect they report on page 8?

Figure S12C displays the allele frequency of the 82R allele across the cell cycle in the single-cell data from allele-replacement strains. These strains were grown separately and processed using two individual 10x chromium runs. The resulting sequenced library had 11,695 cells with the 82R allele and 14,894 cells with the 82W allele. The 7-8% difference in the number of cells is due to slight differences in the number of captured cells per run, not due to growth differences, because we attempted to pool cells in equal numbers from separate mid-log cultures.

The proportion of cells in G1 increases by ~3% in strains with the 82R allele relative to the baseline proportion of cells in the experiment, which, to the reviewers point, is still larger than the ~1% growth difference we observed. Cell cycle occupancy is a compositional phenotype. As shown in figure S12C, the 82R variant increases the fraction of cells in G1 and slightly decreases the fraction of cells in M/G1. There is no obvious expectation for quantitatively translating a change in cell cycle occupancy to a change in growth rate.

The authors refer to the Lang et al. 2009 paper w/respect to GPA1 variant S469I but that paper seems to have explored a different GPA1 allele, GPA1-G1406T, with respect to growth rates.

We thank the reviewer for their comment. The S469I variant is the same as the G1406T variant, one denoting the amino acid change at position 469 in the protein and the other denoting the corresponding nucleotide change at position 1406 in the DNA coding sequence. We have altered the text to make this clear to the reader.

**Reviewer #2 (Recommendations For The Authors):**
I make no recommendations as to additional work for the authors. The manuscript is complete. I suggested some things I would like to see in my review, but it's up to them to decide whether they think any of those would further enhance the manuscript.However, I do have I have some pedantic formatting notes:- Microliters are variously presented as uL, ul, and µl - it should be µL- Similarly, milliliters are presented as ml and ML - it should be mL- Also, there should be a space between the number and the unit, e.g. 10 µL- Some gene names in the manuscript are not italicized in all instances, e.g., GPA1

We thank the reviewer for these formatting suggestions, we have made these changes throughout the text.